# Y-box protein 1 is required to sort microRNAs into exosomes in cells and in a cell-free reaction

Matthew J Shurtleff[1], Morayma M Temoche-Diaz[1], Kate V Karfilis[2], Sayaka Ri[3], Randy Schekman[3]*

[1]Department of Plant and Microbial Biology, University of California, Berkeley, United States; [2]Institute of Molecular Biology, University of Oregon, Eugene, United States; [3]Department of Molecular and Cellular Biology, Howard Hughes Medical Institute, University of California, Berkeley, United States

**Abstract** Exosomes are small vesicles that are secreted from metazoan cells and may convey selected membrane proteins and small RNAs to target cells for the control of cell migration, development and metastasis. To study the mechanisms of RNA packaging into exosomes, we devised a purification scheme based on the membrane marker CD63 to isolate a single exosome species secreted from HEK293T cells. Using immunoisolated CD63-containing exosomes we identified a set of miRNAs that are highly enriched with respect to their cellular levels. To explore the biochemical requirements for exosome biogenesis and RNA packaging, we devised a cell-free reaction that recapitulates the species-selective enclosure of miR-223 in isolated membranes supplemented with cytosol. We found that the RNA-binding protein Y-box protein I (YBX1) binds to and is required for the sorting of miR-223 in the cell-free reaction. Furthermore, YBX1 serves an important role in the secretion of miRNAs in exosomes by HEK293T cells.

*For correspondence:
schekman@berkeley.edu

## Introduction

In contrast to the normal pathways of protein secretion, the processes by which unconventional cargoes are secreted have proved diverse and enigmatic. Indeed, our understanding of unconventional secretory mechanisms is limited to a few examples of leader-less soluble and transmembrane proteins (*Malhotra, 2013*). Unconventionally secreted molecules may be externalized in a soluble form by translocation across various membranes. This may include direct translocation across the plasma membrane, or across an organelle membrane followed by fusion of the organelle with the plasma membrane (*Zhang and Schekman, 2013*). Alternatively, proteins and RNAs can be secreted within vesicles that bud from the plasma membrane, as in the budding of enveloped viruses such as HIV, or within vesicles internalized into a multivesicular body (MVB) that fuses with the plasma membrane (*Colombo et al., 2014*).

RNA is actively secreted into the medium of cultured cells and can be found in all bodily fluids enclosed within vesicles or bound up in ribonucleoprotein complexes, both forms of which are resistant to exogenous ribonuclease (*Colombo et al., 2014*; *Arroyo et al., 2011*; *Mitchell et al., 2008*). Importantly, extracellular vesicle-bound RNAs appear to be enriched in specific classes of RNAs, including small RNAs and microRNA (miRNA) (*Skog et al., 2008*; *Valadi et al., 2007*; *Kosaka et al., 2010*).

Exosomes are a subclass of extracellular vesicle which can be defined as 30–100 nm vesicles with a buoyant density of ~1.10–1.19 g/ml that are enriched in specific biochemical markers, including tetraspanin proteins (*Colombo et al., 2014*). It is often assumed that vesicles fitting this description

**eLife digest** Human cells release molecules into their surroundings via membrane-bound packets called exosomes. These molecules can then circulate throughout the body and are protected from degradation. Among the cargos carried by exosomes are small molecules of RNA known as microRNAs, which are involved in regulating gene activity.

Only a select subset of the hundreds of microRNAs in a human cell end up packaged into exosomes. This suggests that there might be a specific mechanism that sorts those microRNAs that are destined for export. However, few proteins or other factors that might be involved in this sorting process had been identified to date.

Shurtleff et al. set out to identify these factors and started by purifying exosomes from human cells grown in the laboratory and looking for microRNAs that were more abundant in the exosomes than the cells. One exosome-specific microRNA, called miR-223, was further studied via a test-tube based system that uses extracts from cells rather than cells themselves. These experiments confirmed that miR-223 is selectively packed into exosomes that formed in the test tube. Using this system, Shurtleff et al. then isolated a protein called Y-box Protein I (or YBX1 for short) that binds to RNA molecules and found that it was required for this selective packaging.

YBX1 is known to be a constituent of exosomes released from intact cells and may therefore be required to sort other RNA molecules into exosomes. Future studies will explore how YBX1 recognizes those RNA molecules to be exported from cells via exosomes. Also, because exosomes have been implicated in some diseases such as cancer, it will be important to explore what role exosome-specific microRNAs play in both health and disease.

are derived from the multivesicular body, but some evidence suggests that physically and biochemically indistinguishable vesicles bud directly from the plasma membrane (*Booth et al., 2006*). Numerous studies have reported the presence of RNAs, especially miRNAs, from fractions containing exosomes, though many of these studies have relied on isolation techniques (e.g. high speed sedimentation) that do not resolve vesicles from other cellular debris or RNPs (*Bobrie et al., 2012*). Thus, it is difficult to know in which form RNAs are secreted and even more challenging to determine which miRNAs may be specifically secreted as exosome cargo. The use of many different cell lines, bodily fluids and isolation methods to identify which miRNAs are specifically packaged into exosomes further complicates the establishment of widely accepted exosomal miRNA cargo.

Even with the crude preparations that have been characterized, it is clear that RNA profiles from exosomes are distinct from those of the producer cells. Thus RNA capture or stabilization in exosomes is likely to occur through a selective sorting mechanism. RNA packaging may occur by specific interactions with RNA binding proteins that engage the machinery necessary for membrane invagination into the interior of an MVB or by interaction of RNAs with lipid raft microdomains from which exosomes may be derived (*Janas et al., 2015*).

In order to probe the mechanism of exosome biogenesis, we developed procedures to refine the analysis of RNA sorting into exosomes. Using traditional means of membrane fractionation and immunoisolation, we identified unique miRNAs highly enriched in exosomes marked by their content of CD63. This miRNA sorting process was then reproduced with a cell-free reaction reconstituted to measure the packaging of exosome-specific miRNAs into vesicles formed in incubations containing crude membrane and cytosol fractions. Among the requirements for miRNA sorting in vitro, we found one RNA-binding protein, YBX1, which is a known constituent of exosomes secreted from intact cells (*Ung et al., 2014*; *Buschow et al., 2010*).

## Results

### Purified exosomes contain RNA

We first sought to purify exosomes from other extracellular vesicles and contaminating particles containing RNA (e.g. aggregates, ribonucleoprotein complexes) that sediment at high speed. We define exosomes as ~30–100 nm vesicles with a density of 1.08–1.18 g/ml and containing the tetraspanin

protein CD63. Based on these criteria, purified exosomes were recovered using a three-stage purification procedure (*Figure 1a*). First, large contaminating cellular debris was removed during low and medium speed centrifugation and exosomes were concentrated by high-speed sedimentation from conditioned medium. Next, to eliminate non-vesicle contaminants, the high-speed pellet fraction was suspended in 60% sucrose buffer and overlaid with layers of lower concentrations of sucrose buffer followed by centrifugation to float vesicles to an interface between 20 and 40% sucrose. Analysis of this partially purified material by electron microscopy showed vesicles of the expected size and morphology with fewer profiles of larger (>200 nm) membranes and reduced appearance of protein aggregates (*Figure 1b,c* compared to *Figure 1d,e*). Finally, sucrose gradient fractions were mixed with CD63 antibody-immobilized beads to recover vesicles enriched in this exosome marker protein.

To monitor and quantify the exosome purification, we generated a stable, inducible HEK293 cell line expressing a CD63-luciferase fusion. Tetraspanin proteins share a common topology in which the amino- and carboxyl-termini face the cytoplasm resulting in a predicted orientation inside the lumen of an exosomal vesicle. Although an intact C-terminal sequence is reported to be required for the proper localization of CD63 to the cell surface (*Rous et al., 2002*), we found that our overexpressed CD63-luciferase fusion was localized to a variety of cell surface and intracellular membranes (*Figure 1—figure supplement 1*). Using isolated exosome fractions, we confirmed that the CD63-luciferase fusion maintained the expected topology. Luciferase activity was stimulated by the addition of detergent to disrupt the membrane and allow access to the membrane impermeable substrates luciferin and ATP, and to trypsin, which inactivated luciferase activity in the presence but not in the absence of detergent (*Figure 1f*). The CD63-luciferase cell line was then used to monitor exosome purification. CD63-luciferase specific activity increased at each step of the purification, yielding a 5-fold purification of exosomes from the starting 100,000 ×g pellet (*Figure 1g*). Additionally, following immunoisolation, most of the RNA was found in the CD63 positive bound (B) fraction, showing that RNA is associated with purified exosomes from HEK293T cells (*Figure 1f*). These results established that the RNA is associated with CD63-containing exosomes, but not necessarily enclosed within exosomes.

## Exosomes contain selectively packaged miRNAs

Previous reports indicated the presence of miRNAs in fractions containing exosomes but which also contain contaminating particles (*Skog et al., 2008*; *Valadi et al., 2007*; *Bobrie et al., 2012*). To identify the specific miRNAs that are enriched in CD63-positive exosomes from 293T cells, we performed Illumina-based small RNA sequencing on libraries prepared from purified exosomes and from cells. We obtained a total of 123,679 miRNA reads (4.4% of total mapped reads – *Figure 2—source data 1*) in the exosome library representing 502 distinct miRNAs and 880,093 reads (7.3% of total mapped reads – *Figure 2—source data 1*) representing 637 miRNAs in the cell library (*Figure 2a*). To determine if a particular miRNA species was over-represented in exosomes, we analyzed the datasets for reads mapping to miRNA precursors and the targeting or passenger strand of mature miRNAs (*Figure 2b*). Exosomes were slightly enriched in reads mapping to precursor and passenger strand transcripts, however, the vast majority of miRNAs (91% from cells and 88% from exosomes) mapped to the mature targeting strand. The relative abundance of each miRNA was estimated by normalizing to the total number of miRNA-mapped reads (i.e. the number of reads mapped to a miRNA locus divided by the total number of miRNA mapped reads for each dataset – RPM). Of these, 134 and 269 miRNAs were uniquely found in the exosome and cell datasets respectively (*Figure 2a*). Most of the miRNAs uniquely found in exosomes were of very low abundance, with only a few counts for each miRNA. A notable exception was miR-223-3p, which was in the 72nd percentile for normalized reads in exosomes (*Figure 2c,d* – red). The relatively high abundance of miR-223 in exosomes and its low level in the cellular library indicated that miR-223 was very efficiently packaged and secreted via exosomes.

We also identified miRNAs that were found in both libraries but were highly enriched in exosomes. A total of 368 miRNAs were detected in both the exosome and cell libraries. Most of these miRNAs were more enriched in cells than exosomes; however, some (e.g. miR144-3p, miR150-5p, miR142-3p) were highly enriched in exosomes and, like miR223, were moderately abundant in exosomes (*Figure 2c,d* - yellow). To summarize, small RNA sequencing from purified exosomes and subsequent miRNA analysis identified several putative exosomal miRNAs.

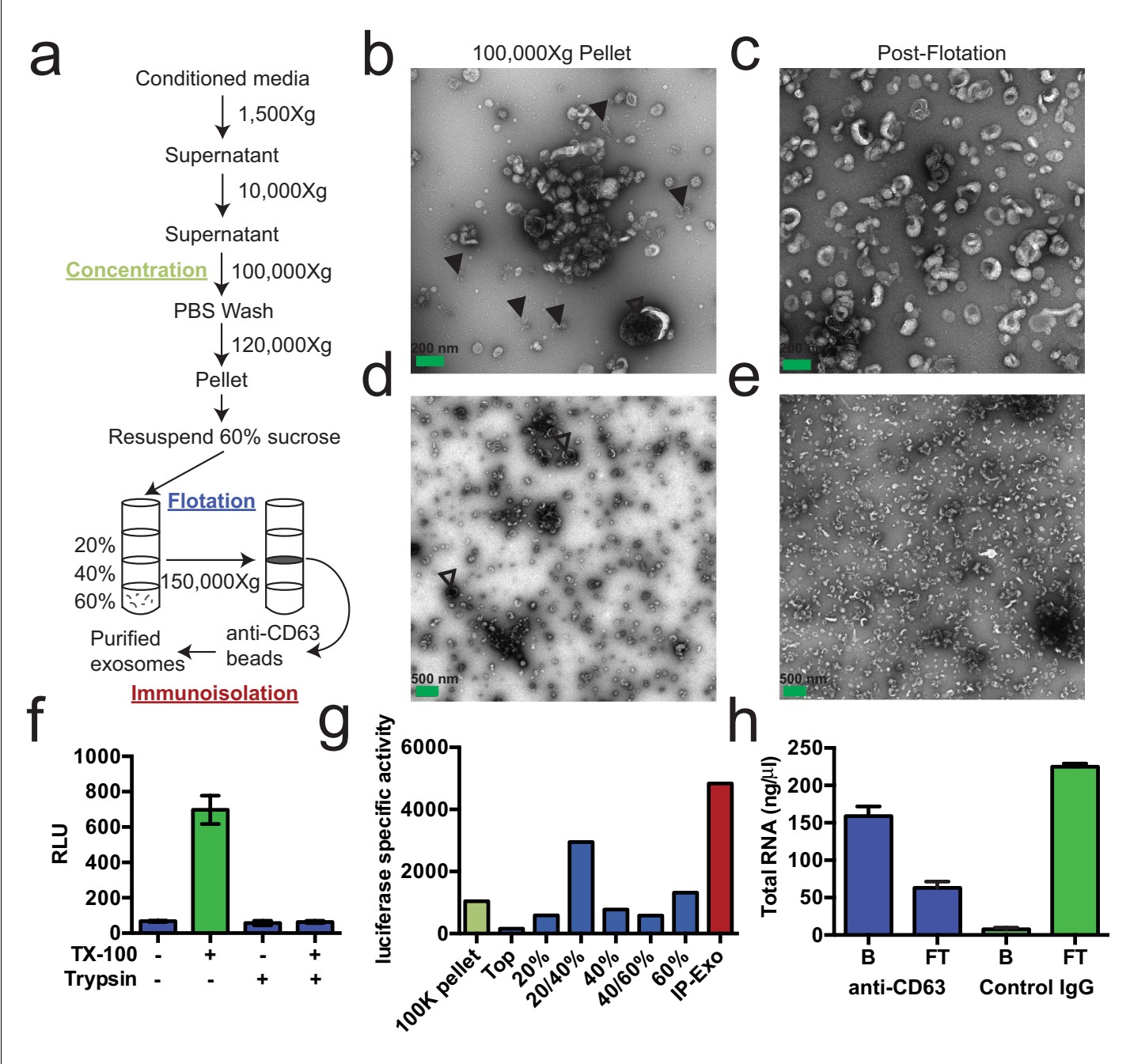

**Figure 1.** Purified CD63-positive exosomes contain RNA. (**a**) Exosome purification schematic. (**b–e**) Representative electron micrographs of negative stained samples from the 100,000 ×g pellet fraction (**b,d**) and post-flotation fractions (**c,e**) at either 9300X (a,b) or 1900X (c,e) magnification. Open arrows indicate large (>200 nm) vesicle contaminants and closed arrows indicate protein aggregates. (**f**) CD63-luciferase activity in purified exosomes after treatment with 1% Triton X-100 (TX-100) and/or 100 μg/ml trypsin for 30 min at 4°C. Error bars represent standard deviations from 3 independent samples. (**g**) Specific activity of CD63-luciferase (RLU/μg of total protein) at each stage of purification (green: 100,000 ×g pellet, purple: post-flotation, red: post-immunoisolation a-CD63 beads). (**h**) Total RNA recovered from conditioned medium after immuno-isolation with a-CD63 or an IgG control. B – bound to beads, FT – flow-through not bound to beads. Error bars represent standard deviations from 3 separate purifications (biological replicates).

The following figure supplement is available for figure 1:

**Figure supplement 1.** Sub-cellular localization of C-terminal CD63-luciferase-FLAG fusion.

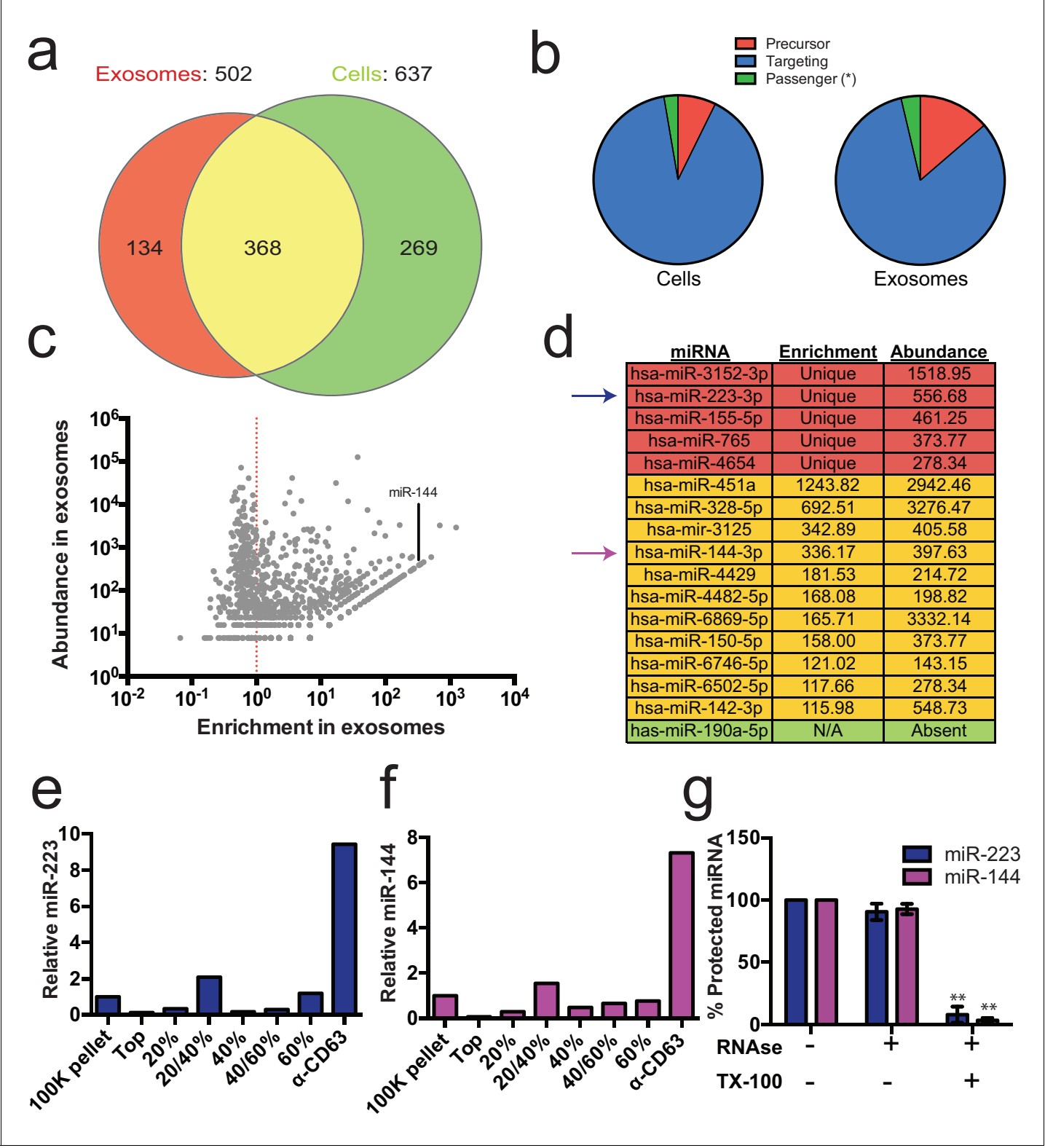

**Figure 2.** Enrichment of select miRNAs in exosomes. (a) Venn diagram showing the number of total (above diagram), unique (inside red or green circles) and shared miRNAs (inside yellow) from each library. (b) Pie charts showing the relative proportion of reads mapping to each miRNA species (Precursor - red, passenger strand – green, targeting strand - blue) in cellular and exosome small RNA libraries (c) Scatterplot showing the enrichment (reads per million miRNA mapped reads (RPM) in exosomes/cells) and relative abundance in exosomes (RPM) of all miRNAs found in both libraries. (d) Table showing the enrichment and abundance (RPM) of relevant miRNAs in exosomes. (Red - unique to exosomes, Yellow - highly enriched in

*Figure 2 continued on next page*

*Figure 2 continued*

exosomes, Green - unique to cells) (e,f) Relative miR-223 (e) and miR-144 (f) per ng of RNA as quantified by qRT-PCR during each stage of the purification. The 100K pellet was set to 1. (g) RNase protection of exosomal miRNAs quantifed by qRT-PCR. Purified exosomes treated with or without RNase I and/or Triton X-100. Errors bars represent the standard deviation from 3 biological replicates. Statistical significance was performed using Student's t-test (**p<0.01).

The following source data and figure supplement are available for figure 2:

**Source data 1.** Mapping statistics for small RNA-seq libraries to the human genome (hg19).
**Source data 2.** miRNA counts from cell and exosomes libraries using miRdeep2.
**Figure supplement 1.** Read frequency distribution along miR-223 and miR-144 precursors.

## miR-223 and miR-144 are selectively packaged exosomal miRNAs

We selected miR-223 and miR-144 from the group of unique and enriched miRNAs for further analysis, as these were the most abundant species with documented functions that mature by the normal pathway of miRNA biogenesis. An analysis of the read frequency distribution of miR-223 and miR-144 from the exosome small RNA-seq dataset showed that the vast majority of reads mapped to the mature guide strand with few reads also mapping to the passenger strand (*Figure 2—figure supplement 1*). We performed quantitative reverse transcriptase PCR (qRT-PCR) for each miRNA target during the course of exosome purification from conditioned medium. Our results showed a selective enrichment of both miR-223 and miR-144 at each stage of the purification (*Figure 2e and f*). Thus, these miRNAs are associated with CD63 exosomes. To determine if these exosome associated RNAs are contained within exosomes and not simply bound to the surface, we performed an RNase protection experiment. Both miRNAs were protected from RNase I digestion, unless detergent was added to disrupt the membrane (*Figure 2g*). These results confirm that miRNAs are selectively packaged into exosomes purified from HEK293T conditioned media and establish miR-223 and miR-144 as specific exosomal miRNAs.

## Cell-free assays for exosome biogenesis and miRNA packaging

### Rationale

The mechanism of exosome biogenesis has been probed in mammalian cell culture using the tools of gene knockdown, knockout and overexpression where it is often difficult to distinguish a primary or indirect role for a gene product. We sought to minimize these challenges by developing simple biochemical assays that reproduce an aspect of exosome biogenesis and miRNA packing in a cell-free reaction.

### Exosome biogenesis in vitro

Since packaging of cargo into newly-formed vesicles presumably occurs concurrently with membrane budding, we sought to develop an assay to monitor the incorporation of an exosome membrane cargo protein into a detergent sensitive membrane formed in an incubation containing membranes and cytsolic proteins obtained from HEK293 cells (*Figure 3a*). Previous studies have reported the use of cell-free assays to monitor multivesicular body biogenesis and sorting of ubiquitinated membrane proteins into intraluminal vesicles (*Tran et al., 2009*; *Falguières et al., 2008*; *Wollert and Hurley, 2010*). To specifically monitor exosome biogenesis, we measured the protection over time of luciferase fused to CD63. The fusion protein displayed luciferase on the cytoplasmic face of a membrane such that its incorporation into a vesicle by budding into the interior of an endosome (or into a vesicle that buds from the cell surface) would render the enzyme sequestered and inaccessible to exogenous luciferin and ATP, the substrates of catalysis (*Figure 1f*). During the incubation, substrate would have access to luciferase exposed on the cytoplasmic face of a membrane or to enzyme about to be internalized into a bud, but not to luciferase that had already become sequestered within vesicles in cells prior to rupture (*Figure 3a*). In order to focus only on luciferase that became sequestered during the cell-free reaction, aliquots taken after a 20 min incubation of membranes,

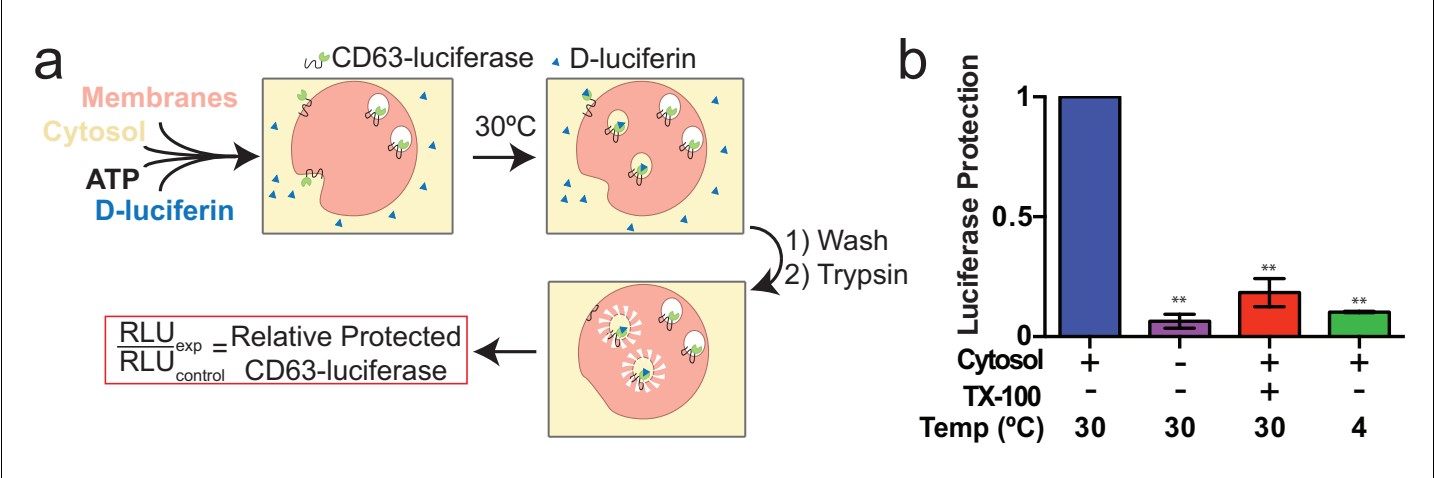

**Figure 3.** Cell-free exosome biogenesis reaction. (a) Schematic illustrating the in vitro biogenesis reaction. (b) Exosome biogenesis measured by relative protected CD63-luciferase. Reactions with or without cytosol, 1%Tx-100 and incubation temperature are indicated. All quantifications represent means from three independent experiments and error bars represent standard deviations. Statistical significance was performed using Student's t-test (**p<0.01).

cytosol and substrate at 30°C were sedimented, resuspended in buffer and sedimented again to remove excess substrate, and CD63-luciferase remaining exposed on the surface of membranes was inactivated by treatment with trypsin (0.5 mg/ml) for 1h at 4°C. Remaining luciferase activity was then monitored in a luminometer. Since the substrates D-luciferin salt and ATP are membrane impermeable, the residual luminescence measured in trypsin-treated samples should derive from luciferase that became segregated along with substrate during the cell-free incubation. Relative protection was quantified as the ratio of relative light units (RLU) for reactions incubated without cytosol, at 4°C and in the presence of detergent (Triton X-100) divided by the RLU for a complete reaction incubated at 30°C. We observed that the formation of sequestered luciferase required cytosol and incubation at 30°C and was disrupted in incubations containing detergent (Triton X-100) (*Figure 3b*). These results suggest that our conditions reconstitute the formation of exosomes in a cell-free reaction.

## MiRNA packaging into vesicles in the cell-free reaction

Having identified miRNAs that are selectively packaged into exosomes in cultured cells, we examined our cell-free reaction for the RNA-selective segregation of miRNAs into RNase resistant and detergent sensitive vesicles (*Figure 4a*). As in the biogenesis reaction described above, crude membranes and cytosol from broken cells were mixed in buffer containing an ATP regenerating system, but in this case supplemented with synthetic miRNA, specifically miR-223, the miRNA that was most highly enriched in exosomes isolated from the medium of cultured HEK293T cells (*Figure 2d*). Given the low relative abundance of miR-223 in HEK293T cells, we set the exogenous concentration in the incubation to be ca. 1000 fold in excess to ensure that the chemically synthetic material predominated in any packaged signal. After a 20 min incubation at 30°C, aliquots were treated with RNase I to digest any unpackaged miRNA. RNA was then purified, reverse transcribed using a miRNA specific primer and the amount of miRNA that became protected during the incubation was measured by quantitative PCR. Packaged RNA was quantified as the percentage of miR-223 RNA molecules protected from RNase during the course of incubation. Packaging of miR-223 required membranes, cytosol and incubation at physiologic temperature (*Figure 4b*). As expected for the segregation of miR-223 into a membrane bound compartment, addition of TX-100 during the RNase incubation abrogated protection (*Figure 4b*). Furthermore, at a minimal concentration of cytosol (0.5 mg/ml), protection was stimulated two-fold over reactions performed in the in the absence of ATP, or in the presence of apyrase or a non-hydrolyzable analog of ATP (*Figure 4c*).

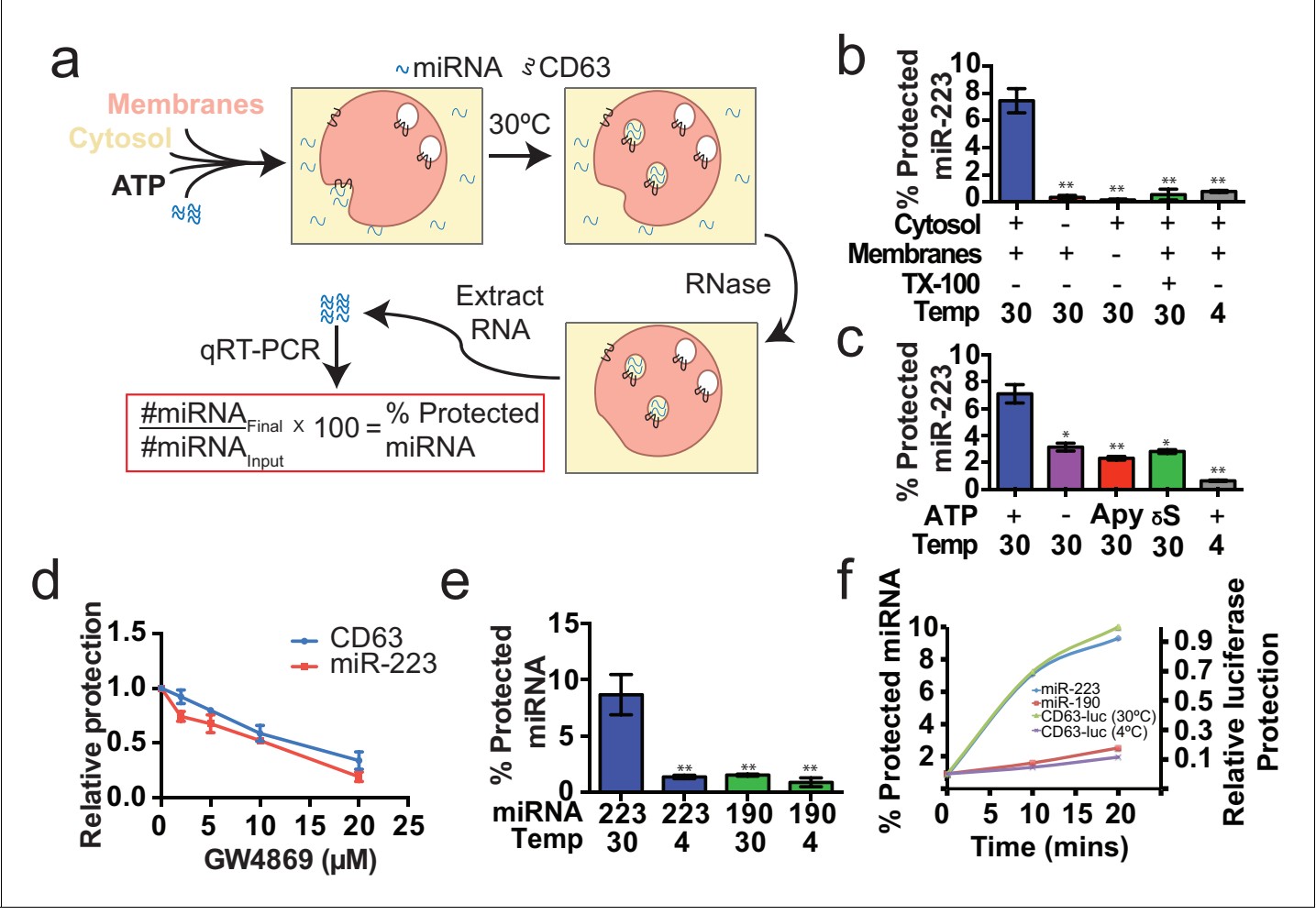

**Figure 4.** Cell-free selective sorting of miRNA into exosomes. (**a**) Schematic illustrating the in vitro packaging reaction. (**b**) Cell-free packaging of miR-223 measured as percent protected by qRT-PCR. Reactions with or without membranes (15,000 ×g pellet), cytosol (100,000 ×g supernatant) and 1% Triton X-100 (TX-100), and incubated at 4 or 30°C are indicated. (**c**) ATP requirements for miR-223 packaging (Apy - Apyrase, (1 U/ml) γS – ATPγS (10 mM)). (**d**) Dose dependent effect of neutral sphingomyelinase 2 inhibitor (GW4869). Measured as relative protection of miRNA and CD63-luciferase normalized to vehicle only control (DMSO). (**e**) Protection of miR-223 or miR-190 measured as a percent protected by qRT-PCR. (**f**) Relative CD63-luciferase (right axis) and percent miRNA protection (left axis) measured over a 20-min time course using the indicated miRNA cargo and incubation temperature. All quantifications represent means from three independent experiments and error bars represent standard deviations. Statistical significance was performed using Student's t-test (*p<0.05, **p<0.01).

To examine the connection between the formation of sequestered luciferase and miR-223 and exosome biogenesis, we performed our cell-free reactions in the presence of an inhibitor (GW4869) of neutral sphingomyelinase, an enzyme that cleaves sphingolipid to form ceramide (*Luberto et al., 2002*). Treatment of cells with an inhibitor of neutral sphingomyelinase 2 (NS2) reduces the secretion of exosomes and exosome-associated miRNAs (*Li et al., 2013*; *Yuyama et al., 2012*; *Trajkovic et al., 2008*). GW4869 inhibited the protection of CD63-luciferase and miR-223 in our cell-free assays at concentrations required to inhibit NS2 activity in partially purified fractions of the enzyme (*Figure 4d*) (*Luberto et al., 2002*). Thus, our cell-free reactions may recapitulate an aspect of exosome biogenesis.

## MiRNA-223 is selectively packaged into vesicles in vitro

Having shown that specific miRNAs are enriched in exosomes produced in vivo, we next determined if selective sorting of miRNAs into vesicles could be reconstituted in vitro. We used the cell-free

packaging assay to compare the efficiency of incorporation of synthetic miR-223 and a relatively abundant cellular miRNA that is not found in exosomes (miR-190a-5p – miR-190). Exosomal miR-223 was more efficiently packaged into vesicles (9%) than cellular miR-190 (1.5%) (*Figure 4e*). Furthermore, the rate of miR-223 packaging mirrored the rate at which luciferase became sequestered in the biogenesis reaction whereas the rate of miR-190 protection in a 30°C incubation reflected the low rate of formation of sequestered luciferase in an incubation held on ice (*Figure 4f*). Based on these experiments, we conclude that the cell-free packaging assay reconstitutes the selective sorting of exosomal miR-223 over cellular miR-190 into vesicles, possibly exosomes, formed in vitro.

## Identifying candidate proteins involved in miRNA sorting into exosomes

To identify proteins that may be involved in miRNA packaging into exosomes, we employed a proteomics approach utilizing the in vitro packaging assay to capture RNA binding proteins. MiRNA sorting may require an RNA binding protein to segregate an RNP into a nascent budded vesicle. Synthetic 3′ biotinylated miR-223 was substituted for unmodified miRNA in the cell-free reaction. Samples were treated with RNase, quenched with RNase inhibitor and solubilized with Triton X-100. miR-223-biotin was captured on streptavidin-coated beads and interacting proteins were eluted with high salt buffer. Mir-223-interacting proteins were identified by in-solution liquid chromatography/ mass spectrometry (*Figure 5a*). Based on peptide count and coverage, the most highly represented protein was Y-box binding protein I (YBX1) (*Figure 5b*). Peptides representing >45% YBX1 of the protein were identified stretching from the cold shock domain to the C-terminus (*Figure 5c*).

YBX1 is a multi-functional RNA binding protein that shuttles between the nucleus, where it plays a role and splice site selection (*Wang et al., 2013*; *Wei et al., 2012*), and the cytoplasm where it is required for the recruitment of RNAs into cytoplasmic ribonucleoprotein granules containing untranslated mRNAs and plays a role in mRNA stability (*Lyabin et al., 2014*). YBX1 also co-localizes with cytoplasmic P-bodies containing members of the RISC complex, including GW182 which can be found in exosomes (*Goodier et al., 2007*; *Gallois-Montbrun et al., 2007*). Interestingly, YBX1 is secreted in a form that resists trypsin in the absence but not in the presence of a non-ionic detergent (Triton X-100), consistent with a location in vesicles, perhaps exosomes (*Frye et al., 2009*; *Rauen et al., 2009*). Furthermore, YBX1 has been detected by mass spectrometry in isolated exosomes (*Ung et al., 2014*; *Buschow et al., 2010*). We first determined if YBX1 co-purifies with exosomes. We purified exosomes as in *Figure 1a* and found that YBX1 was primarily associated with the CD63-bound fraction containing known exosome markers (TSG101, Alix, CD9), as opposed to flotillin 2, which was predominantly found in the unbound fraction (*Figure 5d*). The CD63 positive (exosome) fraction contained most of the RNA (*Figure 1f*). These results show that HEK293T cells release at least two vesicle types (CD63 positive and negative).

We next examined the biochemical requirements for co-packaging of miR-223 and YBX1 using the biotin-miR-223 packaging reaction described in *Figure 5a*. An immunoblot showed YBX1 bound to biotin-mi223 was recovered in exosomes in a complete reaction, while no detectable YBX1 was recovered in exosomes in incubations that lacked cytosol or membranes and much reduced signals in control incubations held at 4C or conducted in the absence of biotin-mi223 (*Figure 5e*). These conditions mirror those required for the packaging of miR-223 in our cell-free reaction.

Because YBX1 is a known RNA binding protein, is secreted by cells in exosomes and physically interacts with miR-223 during the in vitro packaging assay, it met our criteria for a potential exosomal miRNA sorting factor.

## Lack of evidence for a specific role for Ago2 in sorting miR-223 into exosomes

Given that mature miRNA guide sequences in the cell are bound to an argonaute family protein, we were surprised that we did not detect any of these proteins associated with miR-223 in the streptavidin-bound fraction. Argonaute proteins have been variously described as being inside of extracellular vesicles (*Arroyo et al., 2011*; *Turchinovich et al., 2011*) or released as free proteins independent of vesicles (*Squadrito et al., 2014*; *Gibbings et al., 2009*). We first performed immunoblot on 100,000 ×g pellet fractions (which should contain vesicle associated and non-vesicle associated Ago2) and vesicle fractions purified on a buoyant density gradient (which should eliminate

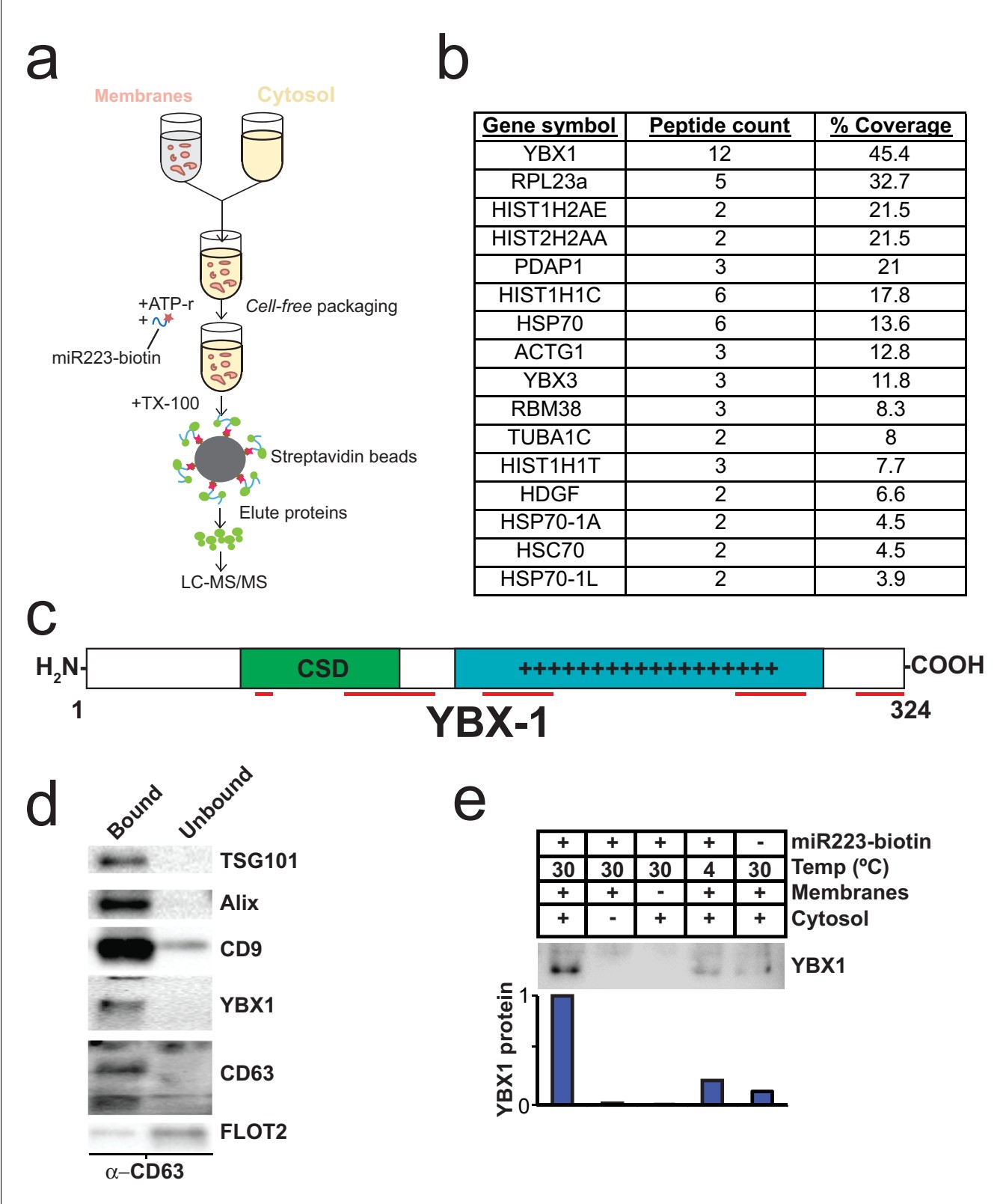

| Gene symbol | Peptide count | % Coverage |
|---|---|---|
| YBX1 | 12 | 45.4 |
| RPL23a | 5 | 32.7 |
| HIST1H2AE | 2 | 21.5 |
| HIST2H2AA | 2 | 21.5 |
| PDAP1 | 3 | 21 |
| HIST1H1C | 6 | 17.8 |
| HSP70 | 6 | 13.6 |
| ACTG1 | 3 | 12.8 |
| YBX3 | 3 | 11.8 |
| RBM38 | 3 | 8.3 |
| TUBA1C | 2 | 8 |
| HIST1H1T | 3 | 7.7 |
| HDGF | 2 | 6.6 |
| HSP70-1A | 2 | 4.5 |
| HSC70 | 2 | 4.5 |
| HSP70-1L | 2 | 3.9 |

**Figure 5.** Identification of YBX1 as a candidate exosomal miRNA sorting protein. (a) Scheme to identify candidate miRNA sorting proteins (b) Proteins identified by tandem mass spectroscopy from the experiment illustrated in (a). (c) Schematic of YBX1 protein. The cold-shock domain (green) and positively charged low-complexity region (blue) are highlighted. Red lines indicate detected unique peptides from mass spectroscopy. (d) Immunoblots for the indicated protein markers in the CD63 immuno-isolated (bound) or unbound fractions. Exosomes were purified as in *Figure 1a*. (e) Immunoblot

*Figure 5 continued on next page*

*Figure 5 continued*

for YBX1 following cell-free packaging reactions performed according to the conditions indicated and immobilized with streptavidin beads as shown in (*Figure 5a*). Bar graph represents densitometry values for the blot shown.

non-vesicle Ago2) to determine if we could detect Ago2 in fractions that contain exosomes. As expected, after flotation the exosome marker proteins were enriched compared to the pellet fraction (*Figure 6a*). In contrast, Ago2 was not detected in the density gradient purified vesicle fraction (*Figure 6a*). The apparent lack of Ago2 in vesicles could be due to its absence in vesicles or a relatively low chemical abundance relative to other molecules. Regardless, our results show that in HEK293T cells the large majority of extracellular Ago2 exists as a non-vesicle associated species. These results support previously published evidence that Ago is not associated with density gradient purified exosomes isolated from breast cancer (MCF7) cells (*Van Deun et al., 2014*).

To further investigate the role of Ago2 in sorting miR-223 into exosomes, we employed the cell-free packaging assay. The inability to detect Ago2 in our mass spectrometry results from in vitro packaged miR-223 could be due to technical limitations of our reaction. We considered two potential technical explanations. First, we used the single stranded guide RNA in our reaction rather than a duplex RNA molecule. Although Ago2 associates with guide RNA in a reaction containing purified components (*MacRae et al., 2008*; *Gagnon et al., 2014*), single-stranded miRNA may be rapidly degraded in a crude extract (*Gagnon et al., 2014*). Thus, it was possible that the duplex molecule might represent a more relevant packaging substrate. Second, 3′ biotinylated RNA was used as the substrate for the packaging reaction. A recent report suggested that, in some cases, 3′ biotinylated RNA could prevent the proper loading of the miRNA into complex with Ago2 (*Guo and Steitz, 2014*). To address these concerns we synthesized miR-223 passenger strand and guide strands biotinylated at the 3′ end or internally at position 13 (*Figure 6b*). The internal position was chosen because in crystal structures of miRNA-protein complexes, middle positions of guide RNAs do not appear to be in direct contact with Ago2 (*Guo and Steitz, 2014*; *Elkayam et al., 2012*; *Schirle and MacRae, 2012*). We then annealed the guide and passenger strands to form the miR-223 duplexes. To determine if the guide or passenger strands can be efficiently loaded into Ago2 in our in vitro reaction conditions, we first mixed the biotinylated substrates with cytosol alone and evaluated complexes that associated with streptavidin beads. In our reaction conditions, both the single stranded guide and duplex oligonucleotides bound apparently equally to Ago2 in cytosol alone and there was no discernible difference in the association comparing 3′ or internally biotinylated oligonucleotides (*Figure 6c*). We then tested the various substrates in our complete packaging reaction including membranes, cytosol and an ATP regenerating system. Duplex substrates were packaged ~2-fold more efficiently than the single stranded guide RNA, irrespective of the position of the biotin group (*Figure 6d*). Interestingly, in reactions programmed with duplex RNA substrate, only the guide RNA and not the passenger RNA was sorted into a protected compartment (*Figure 6d*). In similar incubations, the YBX1 protein was ~2X more efficiently packaged in reactions programmed with duplex RNA but Ago2 was not detected associated with miR-223 in any of the complete reactions (*Figure 6e,f*). This suggests that whereas the RNA substrates are capable of being bound by Ago2 in the cytosol, in the complete reaction containing membranes and ATP, YBX1 is the predominant binding factor. These results explain why Ago2 was not detected in the mass spectrometry data and are consistent with our failure to detect Ago2 in buoyant density-fractionated extracellular vesicles. Similarly, *Van Deun et al. (2014)* found no evidence of Ago2 in density gradient-isolated exosomes from MCF7 cells (*Van Deun et al., 2014*).

## Y-box protein 1 is involved in sorting miR-223 into exosomes

Given the absence of Ago2 in exosomes or associated with miR-223 in our cell-free RNA sorting reaction, we focused on the primary candidate RNA binding protein found in our mass spectrometry results. We next evaluated the requirement for YBX1 in packaging exosomal miRNAs in cells and in the cell-free reaction. To address this question, we generated a YBX1 knockout HEK293T cell line with CRISPR/Cas9 using a guide RNA targeting the YBX1 locus (*Cong et al., 2013*; *Jinek et al., 2013*; *Mali et al., 2013*). Clones were screened by genomic PCR and immunoblot for YBX1. We recovered a homozygous mutant clone (ΔYBX1) that had been targeted at the YBX1 locus and no

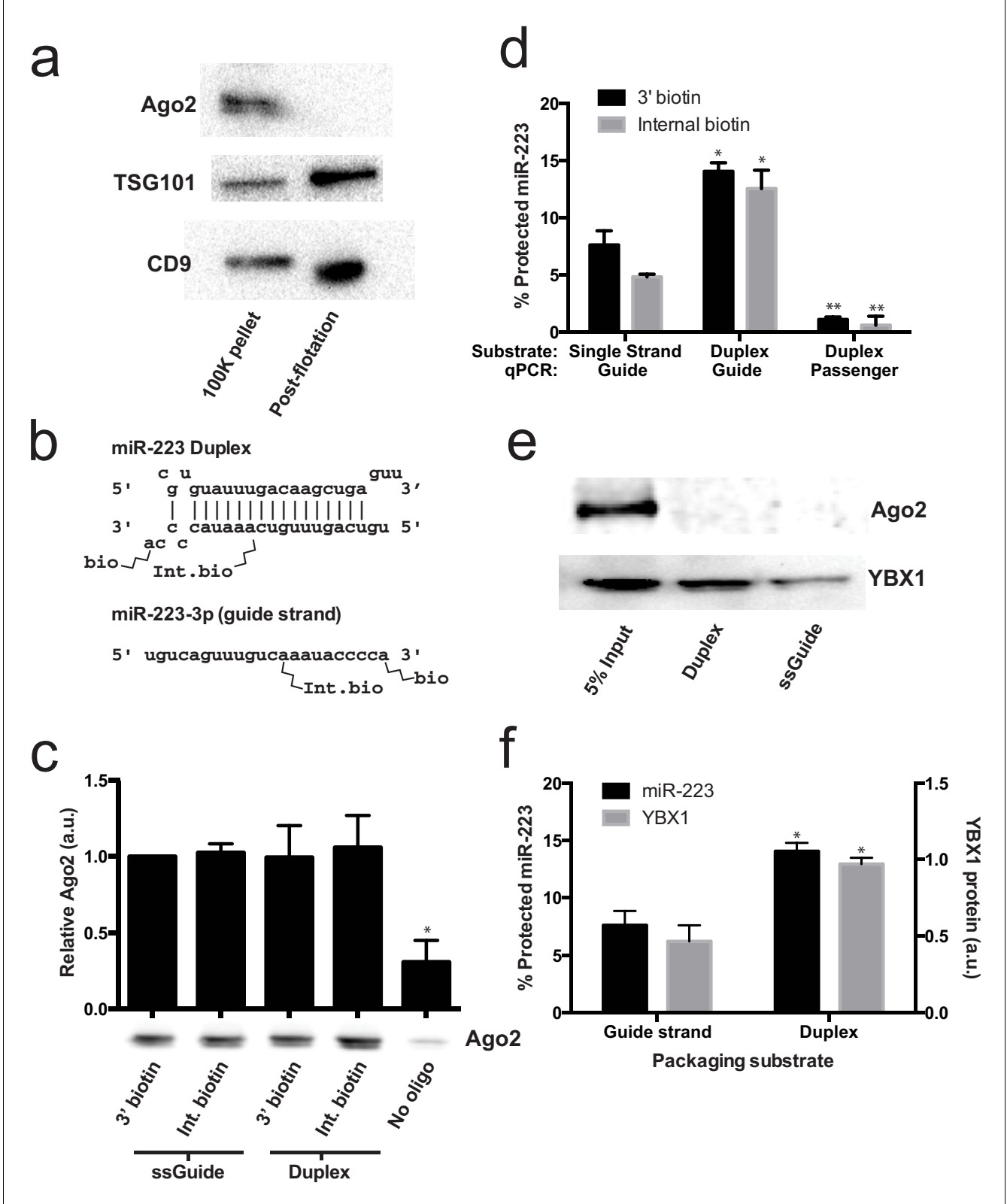

**Figure 6.** Lack of evidence for a specific role for Ago2 in sorting miR-223 into exosomes. (**a**) Immunoblots for Ago2 and exosome markers TSG101 and CD9 in 100,000 ×g (100K) pellet and the 20/40% sucrose interface fractions. (**b**) Schematics showing 3' and internally biotinylated miR-223 duplex and mature guide strand substrates. (**c**) Immunoblots for Ago2 from substrates mixed with cytosol alone for 30 min at 30°C and then absorbed on streptavidin-conjugated beads. (**d**) Percent protected miR-223 (either guide strand or passenger strand) from 3' biotinylated or internally biotinylated

*Figure 6 continued on next page*

Figure 6 continued

single strand or duplex substrates. (e) Immunoblots for Ago2 and YBX1 from substrates packaged in the complete in vitro reaction and then absorbed on streptavidin-conjugated beads. (f) Percent RNAse protected miR-223 and relative level of streptavidin-absorbed YBX1 protein (normalized to duplex). MiR-223 and YBX1 quantification comes from data in d) and e), respectively. All quantifications represent means from 3 independent samples and errors bars represent standard deviations. All quantifications represent means from three independent experiments and error bars represent standard deviations. Statistical significance was performed using Student's t-test (*p<0.05, **p<0.01).

longer expressed YBX1 protein (*Figure 7a*). The homozygous mutant cells grew normally under the

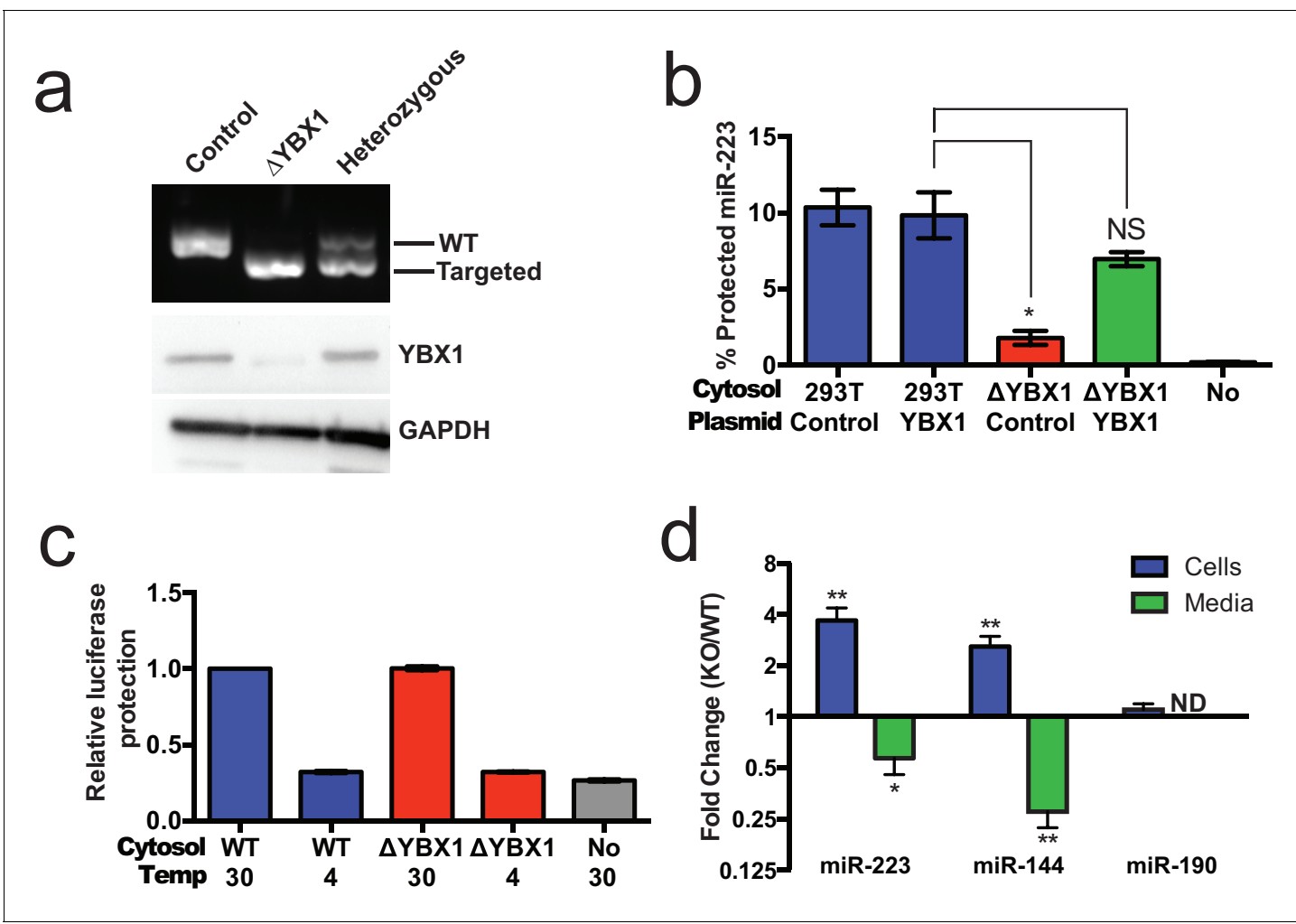

**Figure 7.** YBX1 is necessary for exosomal miRNA packaging and secretion. (a) Analysis of wild-type and CRISPR/Cas9 genome edited HEK293T clones by PCR flanking the genomic target site (top) and immunoblot for YBX1 (middle) and GAPDH (bottom). (b) In vitro miR-223 packaging into exosomes from ΔYBX1 or WT cytosol transfected with control (pCAG) or YBX1 plasmid. (c) Cell-free exosome biogenesis with cytosol from ΔYBX1 or WT cells and membranes from CD63-luciferase cells. (d) Fold change of miR-223 and miR-144 in cells and media from by ΔYBX1 (KO) and WT cells (KO/WT) ND = Not detected. All quantifications represent means from three independent experiments and error bars represent standard deviations. Statistical significance was performed using Student's t-test (*p<0.05, **p<0.01 and NS = not significant).

The following figure supplements are available for figure 7:

**Figure supplement 1.** Partial redundancy for YBX2 for the secretion of miR-223 in cells.

**Figure supplement 2.** miR-223 association with Ago2 in WT and ΔYBX1 cells.

conditions used to propagate HEK293T cells and released an approximately equal number of particles into the medium after 48 hr of growth (2.38 × 10⁷ and 2.42 × 10⁷ particles/ml for wild type and ΔYBX1) as determined by Nanosight nanoparticle tracking analysis.

To determine if YBX1 was required for miRNA packaging, we prepared cytosol from ΔYBX1 cells and tested miR-223 incorporation using the in vitro packaging assay. Cytosol from ΔYBX1 cells did not support miR-223 protection in vitro but activity was largely restored in reactions containing cytosol from a ΔYBX1 line transfected with plasmid encoding YBX1 (*Figure 7b*). We also evaluated the role of YBX1 in our biogenesis reaction (*Figure 3a*) and found that cytosol from wt and ΔYBX1 were indistinguishable in the formation of latent luciferase activity (*Figure 7c*). Thus, YBX1 is required for exosomal miRNA packaging in vitro but is not required for the sorting of an exosome membrane cargo protein into vesicles in our cell-free reaction.

In an effort to connect the results of our cell-free reaction to the mechanism of sorting of miRNAs into exosomes secreted by HEK293T cells, we examined the secretion of miR-223 and of another miRNA, miR-144, which was also highly enriched in our purified exosome fraction (*Figure 2d*). We measured the amount of miR-144 and miR-223 secreted into the medium and retained in cells by qRT-PCR. ΔYBX1 cells showed a significant decrease in secretion of both miRNAs, though more notably of miR-144, during a 24 hr incubation in fresh medium (*Figure 7d*). When the YBX1 paralog YBX2 was knocked down in ΔYBX1 cells, miR-223 secretion was diminished to the baseline level, suggesting partial functional redundancy for YBX paralogs in the secretion of miR-223 (*Figure 7—figure supplement 1*). The defect in miR-223 secretion was less substantial in cells compared to the cell-free reaction (*Figure 7b*), possibly reflecting a rate effect that distinguishes the magnitude of the defect in a 20 min cell-free incubation at 30°C vs. a 24 hr incubation of cells at 37°C. Nonetheless, the secretion defect was accompanied by an ~three–four fold accumulation of each miRNA in cells. No accumulation was observed for another miRNA (miR-190), which is not released in exosomes (*Figure 7d*). To determine if Ago2 binds miR-223 that accumulates in ΔYBX1 cells, we performed RNA immunoprecipitation with Ago2 or isotype matched control IgG in lysates of WT and ΔYBX1 cells and quantified the amount of associated miR-223 by qPCR (*Figure 7—figure supplement 2*). No miR-223 above background co-immunoprecipitated with Ago2 in WT cells. In contrast, miR-223 accumulated in ΔYBX1 cells was found to be associated with Ago2. These results suggest that miR-223 is properly processed in HEK293T cells, likely loaded into Ago2 and then efficiently dissociated and bound by YBX1. Taken together, our results show that YBX1 controls the secretion of select exosomal miRNAs in vitro and in vivo.

## Discussion

Our results establish miR-223 and miR-144 as specific exosome cargo in HEK293T cells. In order to probe the mechanism of RNA sorting into exosomes, we developed biochemical assays that measure the capture of an exosome membrane protein and miRNA into vesicles formed in a cell-free reaction. Using this approach, we identify YBX1 as an RNA-binding protein that is critical for the efficient packaging of miR-223 in vitro and its secretion in cultured human cells.

### How are miRNAs recognized for sorting into exosomes?

We find that synthetic miR-223 is sequestered into vesicles more efficiently than miR-190, consistent with the possibility of a primary RNA sequence or secondary structure, perhaps stabilized by an RNA binding protein such as YBX1, that directs RNA sorting. One possible sorting motif – GGAG – is enriched in miRNAs secreted in exosomes from T-cells (*Villarroya-Beltri et al., 2013*). This motif is recognized by hRNPA2B1, a T-cell exosome RNA-binding protein, which requires sumoylation for efficient secretion via exosomes. It was therefore suggested that binding of GGAG containing miRNAs by sumoylated hRNPA2B1 was a sorting mechanism for miRNAs into T-cell-derived exosomes.

We were unable to identify any statistically significant primary sequence motifs for miRNAs by either multiple alignment (ClustalW) or multiple Em for motif elicitation (MEME) in HEK293T-derived exosomes (*Larkin et al., 2007*; *Bailey et al., 2009*). Furthermore, the mature targeting strand of miR-223 packaged into exosomes contains no guanine nucleotides and hRNPA2B1 was not detected in our mass spectrometry results for proteins bound to miR-223-biotin isolated from vesicles formed in our cell-free reaction. The human genome encodes more than 1000 experimentally determined and predicted RNA-binding proteins (*Cook et al., 2011*; *Gerstberger et al., 2014*). We therefore

propose that different cell types may use RNA binding proteins with distinct binding preferences to secrete miRNAs, and perhaps other RNA classes, in exosomes. In addition, some cell types may deploy multiple RNA-binding proteins to sort RNAs into exosomes, in which case motif discovery would be challenging, even in highly purified vesicles, due to diverse motif preferences from distinct proteins.

We identified YBX1 as the dominant RNA-binding protein physically interacting with miR-223 in vitro and confirmed its role in miR-223 packaging into exosomes both in vitro and in cultured cells. YBX1 is found within mammalian P-bodies (GW bodies) containing untranslated RNAs (*Kedersha and Anderson, 2007*) and in the nucleus where it plays a role in RNA splice site selection by binding short sequence motifs (*Wang et al., 2013*; *Wei et al., 2012*). YBX1 binds RNA via an internal cold shock domain and an inherently disordered, highly charged C-terminus (*Lyabin et al., 2014*). Interestingly, another cold shock domain containing protein, Lin28, binds pre-miRNAs of the Let7 family via hairpin-loop structures (*Nam et al., 2011*). YBX1 also binds hairpin-loops in a murine retrovirus, leading to stabilization of the viral RNA genome and increased particle production (*Bann et al., 2014*). YBX1 binding of viral RNA also increases production of other retroviruses, including HIV (*Bann et al., 2014*; *Mu et al., 2013*; *Li et al., 2012*). This raises the possibility that the recognition motif for sorting into exosomes may be based on secondary rather than primary RNA structure and that YBX1 may act as an RNA co-factor to escort exosomal RNAs into exosomes.

Our studies focused on miRNAs, however it is possible that YBX1 is responsible for the secretion of other RNA classes in exosomes. Several recent reports indicate a role for YBX1 in binding various small RNAs, including miRNAs, tRNA fragments and snoRNAs (*Blenkiron et al., 2013*; *Liu et al., 2015*; *Goodarzi et al., 2015*). It is interesting to note that most miRNAs present in exosomes in our study are not highly enriched compared to their relative abundance in cells. This raises the possibility that highly enriched exosomal miRNAs mimic other classes of RNAs that are more efficiently packaged in a YBX1-dependent manner.

## Chaperone-mediated sorting of miRNAs into exosomes

A surprising finding from our study is the lack of evidence for the argonaute proteins in isolated exosomes or sequestered with miR-223 in our cell-free RNA sorting reaction. This in spite of our observation that cell-free reaction reconstitutes sorting of the mature strand from a duplex RNA. Some recent evidence suggests that Ago2 may be sorted along with miRNAs into exosomes as a result of aberrant KRAS signaling. Analysis of extracellular RNA in isogenic cell lines differing only in KRAS status revealed that secretion of a sub-population of miRNAs is decreased in colorectal cancer cells harboring an activating KRAS mutation whereas other miRNAs are secreted at equivalent levels irrespective of KRAS status (*Cha et al., 2015*). Subsequent experiments showed that KRAS mutation results in phosphorylation of Ago2 causing its re-localization from multivesicular bodies to P-bodies leading to decreased secretion of select miRNAs (*McKenzie et al., 2016*). Notably, the miRNA (miR-223-3p) shown here to be dependent on YBX1 is among the cohort of miRNAs that were not affected by KRAS status. These results combined with ours suggest two possible routes for miRNA egress via exosomes, an Ago2-associated pathway and a RNA-binding protein-dependent pathway that we term chaperone-mediated sorting. The chaperone-mediated pathway would include the previously identified hnRNPA2B1 in T-cells (*Villarroya-Beltri et al., 2013*) and YBX1 in HEK293T cells. Interestingly, one other highly enriched miRNA (miR-328-5p) has been previously shown to conditionally associate with either Ago2 or with the RNA-binding protein hnRNPE2 (*Eiring et al., 2010*), suggesting that these two pathways may not be mutually exclusive. A notable feature of the chaperone-mediated pathway is that the RNA content of exosomes may be manipulated by altering the expression of individual RNA binding proteins involved in RNA export with distinct nucleic acid binding specificities. Further characterization of the chaperone-mediated pathway may then allow for targeted sorting of engineered RNA species into exosomes.

## The physiological role of exosomal miRNA

The functional role of secreted miRNAs has been a matter of discussion since the first reports of extracellular RNA (*Théry, 2011*). Numerous studies have shown that miRNAs can be transferred to neighboring cells in experimental settings (*Skog et al., 2008*; *Valadi et al., 2007*; *Kosaka et al., 2010*; *Cha et al., 2015*; *Pegtel et al., 2010*; *Mittelbrunn, 2011*; *Rana et al., 2013*). However, the

transfer of miRNAs in biologically significant quantities for function in a physiological context is far from proven. Indeed, a recent study reported a stoichiometry of less than one specific miRNA per exosome, with the caveat that this study characterized crude, high-speed pellet fractions from conditioned medium (*Chevillet et al., 2014*). Functional miR-223 transferred between macrophages and miR-223-containing exosomes can induce macrophage differentiation, however, it has yet to be shown that miR-223 transfer plays a direct role in the differentiation (*Ismail et al., 2013*). Indeed, direct and convincing evidence for a physiological role of miRNAs secreted via exosomes has so far proven elusive. Alternatively, exosomes may be a convenient carrier to purge unnecessary or inhibitory RNAs from cells. A recent report provided evidence for both alternative views with the demonstration that target transcript levels for miRNAs in the cell modulate the abundance of miRNAs in macrophage exosomes, and this in turn dictates which miRNAs are transferred to repress transcripts in recipient cells (*Squadrito et al., 2014*). Because YBX1 and the RISC machinery have both been shown to localize to P-bodies and P-bodies are closely juxtaposed to multivesicular bodies, all of the necessary machinery is poised to efficiently secrete miRNAs in exosomes (*Gibbings et al., 2009*). YBX1 may complex with miRNAs whose mRNA targets are not expressed, and sort them into the intralumenal vesicles of a multivesicular body for export by unconventional secretion. Physiological studies of the function miRNAs that are secreted via the Ago2-associated vs. chaperone-mediated pathways may explain contradictory results for different miRNAs and provide general rules for extracellular miRNA function.

## Materials and methods

### Cell lines, media and general chemicals

HEK293T cells were cultured in DMEM with 10% FBS (Thermo Fisher Scientific, Waltham, MA). HEK293T cell lines were maintained by the UC-Berkeley Cell Culture Facility and were confirmed by short tandem repeat profiling (STR) and tested negative for mycoplasma contamination. For exosome production, cells were seeded to ~10% confluency in 150 mm CellBIND tissue culture dishes (Corning, Corning NY) containing 30 ml of growth medium and grown to 80% confluency (~48 hr). We noted that confluency >80% decreased the yield of exosome RNA. Cells grown for exosome production were incubated in exosome-free medium produced by ultracentrifugation at 100,000 ×g (28,000 RPM) for 18 hr using an SW-28 rotor (Beckman Coulter, Brea, CA) in a LE-80 ultracentrifuge (Beckman Coulter). Unless otherwise noted, all chemicals were purchased from Sigma Aldrich (St. Louis, MO).

### Exosome purification

Conditioned medium (3 l for small RNA-seq and 420 ml for all other experiments) was harvested from 80% confluent HEK293T cultured cells. All subsequent manipulations were performed at 4°C. Cells and large debris were removed by centrifugation in a Sorvall R6+ centrifuge (Thermo Fisher Scientific) at 1500 ×g for 20 min followed by 10,000 ×g for 30 min in 500 ml vessels using a fixed angle FIBERlite F14-6X500y rotor (Thermo Fisher Scientific). The supernatant fraction was then passed through a 0.22 μM polystyrene vacuum filter (Corning) and centrifuged at ~100,000 ×g (26,500 RPM) for 1.5 hr using two SW-28 rotors. The maximum rotor capacity was 210 ml, thus the small RNA-seq processing required pooling from ~15 independent centrifugations. The pellet material was resuspended by adding 500 μl of phosphate buffered saline, pH 7.4 (PBS) to the pellet of each tube followed by trituration using a large bore pipette over a 30 min period at 4°C. The resuspended material was washed with ~5 ml of PBS and centrifuged at ~120,000 ×g (36,500 RPM) in an SW-55 rotor (Beckman Coulter). Washed pellet material was then resuspended in 200 μl PBS as in the first centrifugation step and 1 ml of 60% sucrose buffer (20 mM Tris-HCl, pH 7.4, 137 mM NaCl) was added and mixed with the use of a vortex to mix the sample evenly. The sucrose concentration in the PBS/sucrose mixture was measured by refractometry and, if necessary, additional 60% sucrose buffer as added until the concentration was >50%. Aliquots (1 ml) of 40%, 20% and 0% sucrose buffer were sequentially overlaid and the tubes were centrifuged at ~150,000 ×g (38,500 RPM) for 16 hr in an SW-55 rotor. The 20/40% interface was harvested, diluted 1:5 with phosphate buffered saline (pH 7.4) and 1 μg of rabbit polyclonal anti-CD63 H-193 (Santa Cruz Biotechnology, Dallas, TX) was added per liter of original conditioned medium and mixed by rotation for 2 hr at 4°C. Magvigen

protein-A/G conjugated magnetic beads (Nvigen, Sunnyvale, CA) were then added to the exosome/antibody mixture and mixed by rotation for 2 hr at 4°C. Beads with bound exosomes were washed three times in 1 ml PBS and RNA was extracted using Direct-Zol RNA mini-prep (Zymo Research, Irvine, CA) or protein was extracted in 100 µl 1X Laemmli sample buffer and dispersed with the use of a vortex mixer for 2 min.

## Negative staining and visualization of exosomes by electron microscopy

An aliquot (4 µl) of the resuspended 100,000 ×g pellet fraction or a sample from the 20/40% interface that was diluted 10-fold with PBS, centrifuged at 100,000 ×g in a TLS-55 rotor and then resuspended in 1% glutaraldehyde, was spread onto glow discharged Formvar-coated copper mesh grids (Electron Microscopy Sciences, Hatfield, PA) and stained with 2% Uranyl acetate for 2 min. Excess staining solution was blotted off with filter paper. Post drying, grids were imaged at at 120 kV using a Tecnai 12 Transmission Electron Microscope (FEI, Hillsboro, OR) housed in the Electron Microscopy Laboratory at UC Berkeley.

## Nanoparticle tracking analysis

Conditioned medium (1 ml) from wild type and ΔYBX1 cells was harvested and the supernatant from a 10,000 ×g centrifugation was drawn into a 1 ml syringe and inserted into a Nanosight LM10 instrument (Malvern, UK). Particles were tracked for 60 s using Nanosight nanoparticle tracking analysis software. Each sample was analyzed 4 times and the counts were averaged.

## Construction of inducible 293:CD63-luciferase cell line and luciferase activity assays

HEK293 cells expressing doxycycline-inducible CD63-luciferase was generated using the T-REx - 293 cell line according to the manufacturer's instructions (Life Technologies, Grand Island, NY). The open reading frame for CD63-was amplified from human cell cDNA and firefly luciferase-FLAG was amplified from a plasmid source, both using Phusion DNA Polymerase (NEB). CD63 was fused to luciferase by NotI digestion, ligation and PCR amplification. The CD63-luciferase-FLAG amplicon was then digested and ligated into pcDNA5/FRT/TO (Life Technologies) using NdeI and PstI sites. The resulting plasmid was co-transfected with pOG44 (Life Technologies) and a stable cell line was selected using hygromycin selection (100 µg/ml). CD63-luciferase expression was induced with 1 µg/ml doxycycline 48 hr prior to exosome harvesting. Luciferase activity was measured using a Promega Glowmax 20/20 luminometer (Promega, Madison, WI) with a signal collection integration time of 1 s. Luciferase reactions contained 50 µl sample, 10 µl 20X luciferase reaction buffer (500 mM Tricine, pH 7.8, 100 mM $MgSO_4$, 2 mM EDTA), 10 µl 10 mM D-luciferin dissolved in PBS, 10 µl ATP dissolved in deionized water and 120 µl deionized water. Where indicated, samples were pre-treated with final concentrations of 1% Triton X-100 and/or 100 µg/ml trypsin for 30 min on ice. Total protein concentrations were measured using Pierce BCA protein assay according to the manufacturer's instructions.

## Immunoblotting

Exosome and cell lysates were prepared by mixing in lysis buffer (10 mM Tris-HCl, pH 7.4, 100 mM NaCl, 0.1% sodium dodecyl sulfate, 0.5% sodium deoxycholate, 1% Triton X-100, 10% glycerol). Lysates were diluted four-fold with 4X Laemmli sample buffer, heated to 65°C for 5 min and separated on 4–20% acrylamide Tris-Glycine gradient gels (Life Technologies). Proteins were transferred to polyvinylidene difluoride membranes (EMD Millipore, Darmstadt, Germany), blocked with 5% bovine serum albumin in TBST and incubated overnight with primary antibodies. Blots were then washed with TBST, incubated with anti-Rabbit or anti-Mouse secondary antibodies (GE Healthcare Life Sciences, Pittsbugh, PA) and detected with ECL-2 reagent (Thermo Fisher Scientific). Primary antibodies used in this study were anti-YBX1 (Cell Signaling Technology, Danvers, MA), anti-GAPDH (Santa Cruz Biotechnology), anti-TSG101 (Genetex, Irvine, CA), anti-CD9 (Santa Cruz Biotechnology), anti-Flotillin 2 (Abcam, Cambridge, MA), anti-Alix (Santa Cruz Biotechnology) and anti-Ago2 (Cell Signaling Technology). For quantitative immunoblotting (in *Figure 6*), the same procedures were used, but were instead imaged using the LiCOR Odyssey imaging system.

## Quantitative real-time PCR

RNA was extracted using the Direct-Zol RNA mini-prep and cDNA was synthesized either by oligo-dT priming (mRNA) or gene-specific priming (miRNA) according to the manufacturer's instructions. For miRNA, we used Taqman miRNA assays from Life Technologies (assay numbers: hsa-mir-223-3p: 000526, hsa-mir-190a-5p: 000489 and hsa-miR-144-3p: 002676). Because there is no well-accepted endogenous control transcript for exosomes, relative quantification was performed from equal amounts of total RNA. Qubit (Thermo Fisher Scientific), was used to quantify total RNA from the medium or cells: 10 ng of RNA was reverse transcribed and qPCR was performed according to manufacturer's instructions. Relative quantification was calculated from the expression $2^{-(Ct_{control}-Ct_{experimental})}$. Taqman qPCR master mix with no amperase UNG was obtained from Life Technologies and quantitative real-time PCR was performed using an ABI-7900 real-time PCR system (Life Technologies).

## Cell-free biochemical assays

### Preparing membranes and cytosol

HEK293T cells were harvested at ~80% confluency by gently pipetting with PBS. Cells were centrifuged at 500 ×g and the pellet was weighed and frozen at −80°C until use. Cells were thawed, resuspended in 2 vol of homogenization buffer (250 mM sorbitol, Tris-HCl, pH 7.4, 137 mM NaCl) containing protease inhibitor cocktail (1mM 4-aminobenzamidine dihydrochloride, 1 µg/ml antipain dihydrochloride, 1 µg/ml aprotinin, 1 µg/ml leupeptin, 1 µg/ml chymostatin, 1 mM phenymethylsulfonly fluoride, 50 µM N-tosyl-L-phenylalanine chloromethyl ketone and 1 µg/ml pepstatin) and passed 7–15 times through a 22 gauge needle until >80% of cells were disrupted, as assessed by microscopy and trypan blue staining. The homogenized cells were then centrifuged at 1500 ×g and the supernatant fraction was centrifuged at 15,000 ×g using a FA-45-30-11 rotor and Eppendorf 5430 R centrifuge (Eppendorf, Hamburg, Germany). The supernatant fraction was centrifuged again at 55,000 RPM in a TLS-55 rotor and Optima Max XP ultracentrifuge (Beckman Coulter) to generate the cytosol fraction (~5 mg/ml). The 15,000 ×g pellet fraction was resuspended in 2 packed cell vol homogenization buffer and an equal vol of 1 M LiCl. The membranes were then centrifuged again at 15,000 ×g and resuspended in 1 original packed cell vol to generate the membrane fraction.

### Cell-free exosome biogenesis assay

Membranes were prepared from HEK293:CD63-luciferase cells and cytosol from HEK293T cells. Complete biogenesis reactions (40 µl) consisted of 10 µl membranes, 17 µl cytosol (2 mg/ml final concentration) + homogenization buffer, 4 µl 10X ATP regeneration system (10 mM ATP, 500 mM GDP-mannose, 400 mM creatine phosphate, 2mg/ml creatine phosphokinase, 20 mM HEPES, pH 7.2, 250 mM sorbitol, 150 mM KOAc, 5 mM MgOAc), 8 µl 5X incorporation buffer (80 mM KCl, 20 mM CaCl2, 12.5 mM HEPES-NaOH, pH 7.4, 1.5 mM, MgOAc, 1 mM DTT), 1 µl D-luciferin (10 mM in PBS). The reaction mixture was incubated at 30°C for 20 min and membranes were sedimented at 15,000 ×g at 4°C. Post-reaction membranes were then resuspended in 500 µl PBS with 0.5 mg/ml trypsin and incubated at 4°C for 1 hr to inactivate CD63-luciferase that had not been internalized during the incubation period. The trypsin-treated reactions were then incubated for 2 min at 25°C and luciferase activity was quantified using the luminometer conditions described above. Exosome biogenesis for experimental conditions was calculated as the relative ratio compared to the complete control reaction described above ($RLU_{experimental}/RLU_{control}$).

### Cell-free exosome miRNA packaging assay

RNA oligonucleotides corresponding to the targeting or passenger strand sequences of miR-223-3p or the targeting strand of miR-190a-5p were purchased from Integrated DNA Technologies (IDT, Coralville, IA). Duplex substrates were generated by incubating mature and targeting strands dissolved in annealing buffer (10 mM Tris, pH 7.5, 20 mM NaCl in RNase-free water) at 95°C for 2 min in a heat block and then removing the block from heat and allowing the samples to cool to room temperature over the course of 1–2 hr on the bench top (*Tuschl et al., 1999*).

Membranes and cytosol were prepared from HEK293T cells. Complete miRNA packaging assays (40 µl) contained 10 µl membranes, ~16 µl cytosol + homogenization bufffer (2 mg/ml final concentration), 4 µl 10X ATP regenerating system, 8 µl incubation buffer, 1 µl 10 nM synthetic single

stranded or duplex miRNA and 1 µl RNAsin (Promega). Reactions were incubated for 20 min at 30°C then placed on ice and mixed with 4.3 µl of 10X NEB buffer 3 and 1 µl of RNase $I_f$ (50,000 units/ml) NEB, Ipswich, MA) was added to all reactions except to a no RNase control. Reactions were then incubated at 30°C for a further 20 min. Following incubation, RNA was immediately extracted according to Direct-Zol (Zymo Research) manufacturer's instructions. First strand complementary DNA synthesis and quantitative PCR was performed using TaqMan miRNA assays (Life Technologies) for hsa-miR-223 or hsa-miR-190. Percent protection was calculated from the qPCR data by comparing the Ct of miRNA in the RNase treated samples against the no RNase control reaction ($2^{-(Ct_{experimental}-Ct_{control})}$)) in which the no RNase control was set to 100 percent.

## Streptavidin pull-down of miR-223 and interacting proteins

The in vitro packaging assay was performed as described above with miR-223-3p with biotin linked either to the 3' phosphate or internally biotinylated at position 13 (IDT). Samples were heated to 65°C for 20 min to inactivate RNase $I_f$ and then mixed with 4.4 µl 10% Triton X-100 for a final concentration of 1% and kept on ice for 30 min. Novagen MagPrep Streptavidin-coated beads (10 µl/reaction) (EMD Millipore) were washed 3 times with 1 ml PBS and then added to the reaction lysate. The suspension was mixed by rotation for 2 hr at 4°C, the beads were immobilized using a magnet and washed 3 times with 1 ml PBS. Proteins were eluted from bead-bound miR-223 with 50 µl 1 M KCl. In-solution liquid chromatography and mass spectrometry were performed according to standard procedures by the Vincent J. Coates Proteomics/Mass Spectroscopy laboratory (UC Berkeley).

## Small RNA sequencing of cellular and exosomal RNA

RNA was prepared from cells and 3 l of HEK293T conditioned media. Sequencing libraries were generated using the Scriptminer Small RNA sequencing kit (Epicentre Biotechnologies, Madison, WI) from 1 µg total RNA from cells and 200 ng total RNA from exosomes according to the manufacturer's protocol. The libraries were amplified and index barcodes were added by 11 cycles of PCR. Libraries were sequenced by 50 bp single read massively parallel sequencing on an Illumina Hi-Seq 2000 System at the Vincent J. Coates Genomic Sequencing Laboratory (UC Berkeley).

## Small RNA sequencing analysis

Preprocessing of the 50 base pair single reads was filtered for read quality (read quality >20 and percent bases in sequence that must have quality >90) and adaptor sequences were clipped using the FASTX toolkit (http://hannonlab.cshl.edu/fastx_toolkit/) implementation on the GALAXY platform (usegalaxy.org) (*Blankenberg, 2010*; *Goecks et al., 2010*; *Giardine et al., 2005*). Sequences were mapped to miRbase using miRdeep2 and counts tables were obtained using the quantifier program using default settings (*Friedländer et al., 2012*). Reads were normalized by dividing the number of reads mapping to each miRNA by the number of total reads mapping to all miRNAs and the quotient was then multiplied by one million (reads per million miRNA mapped reads - RPM). To analyze miRNA species, we used the quantifier program of the miRdeep2 software suite (*Friedländer et al., 2012*). Precursor reads were determined by subtracting the number of reads mapping to mature (either targeting or passenger strand sequences) from the total number of reads mapping to the full-length precursor transcript for each miRNA. Those miRNAs with described passenger strands (star strand) were then analyzed to determine how many mature reads mapped to either targeting or passenger strands.

## CRISPR/Cas9 genome editing

A pX330-based plasmid expressing venus fluorescent protein was kindly provided by Robert Tjian (*Cong et al., 2013*). A CRISPR guide RNA targeting the first exon of the YBX1 open reading frame was selected using the CRISPR design tool (*Hsu et al., 2013*). The YBX1 guide RNA was introduced into pX330-Venus by oligonucleotide cloning as described (*Cong et al., 2013*). HEK293T cells were transfected for 48 hr at low passage number, trypsinized and sorted for single, venus positive cells in a 96 well plate using a BD Influx cell sorter. Wells containing single clones (16 clones) were allowed to expand and were screened by semi-nested PCR using primers targeting the genomic region flanking the guide RNA site. Primers for the first round of PCR (10 cycles) were: YBX1-F1 (GGTTGTAGG TCGACTGAATTA) and YBX1-R1 (ACCGATGACCTTCTTGTCC). The PCR primers from the first

round were removed using DNA clean and concentrator-5 kit (Zymo Research) according to manufacturer's instructions and the second round of PCR (25 cycles) was performed with primers: YBX1-F2 (CGGCCTAGTTACCATCACA) and YBX1-R1 (ACCGATGACCTTCTTGTCC). PCR products were separated on a 2.5% agarose gel to identify products smaller than the wild type PCR product, indicating a deletion. Clones (8) showing homozygous or heterozygous mutations were then screened by immunoblot to identify those that did not express YBX1. A clone containing a single homozygous mutation at the target site and not expressing YBX1 by immunoblot was recovered and designated ΔYBX1.

### siRNA Knockdown and measurement of miR-223 secretion

Predesigned siRNA oligos targeting YBX2 were obtained from Qiagen (Hs_YBX2_3:AAGCCGGTGCTGGCAATCCA). Cells were seeded at 60% confluency and siRNA was transfected using Lipofectamine 2000 reagent (Life Technologies). After 48 hr the media was replaced with exosome depleted media and allowed to incubate another 24 hr. An aliquot (1 ml) of medium was removed and centrifuged at 1500 and 15,000 ×g and then extracted using the Zymo RNA Prep Kit. 1 ng of RNA was reversed transcribed and qPCR was performed as described above.

### RNA immunoprecipitation

RNA immunoprecipitation was performed using the Magna RIP kit (Millipore) according to the manufacturer's instructions. Mouse monoclonal anti-Ago antibody (5 μg, clone 2A8) purchased from Sigma-Aldrich was used to immunoprecipitate from a lysate generated from $2.0 \times 10^7$ HEK293T cells (WT or ΔYBX1) in 100 μl Magna RIP lysis buffer.

## Acknowledgements

We thank Amita Gorur for help and advice on electron microscopy. We also thank the staff at the UC-Berkeley shared facilities used in this work, the Functional Genomics Laboratory, the Vincent J Coates Sequencing Facility, the Vincent J Coates Proteomics Facility, the Computational Genomics Resource Laboratory and the Flow Cytometry Facility. This work used the Vincent J Coates Genomics Sequencing Laboratory at UC Berkeley, supported by NIH S10 Instrumentation Grants S10RR029668 and S10RR027303 and the Vincent J Proteomics/Mass Spectrometry Laboratory at UC Berkeley, supported in part by NIH S10 Instrumentation Grant S10RR025622.

## Additional information

### Competing interests

RS: Editor in Chief, *eLife*. The other authors declare that no competing interests exist.

### Funding

| Funder | Grant reference number | Author |
| --- | --- | --- |
| Howard Hughes Medical Institute | Investigator | Randy Schekman |

The funders had no role in study design, data collection and interpretation, or the decision to submit the work for publication.

### Author contributions

MJS, Conception and design, Acquisition of data, Analysis and interpretation of data, Drafting or revising the article; MMT-D, Acquisition of data, Analysis and interpretation of data, Drafting or revising the article; KVK, Analysis and interpretation of data, Drafting or revising the article; SR, Acquisition of data, Analysis and interpretation of data; RS, Conception and design, Analysis and interpretation of data, Drafting or revising the article

### Author ORCIDs

Matthew J Shurtleff, http://orcid.org/0000-0001-9846-3051
Randy Schekman, http://orcid.org/0000-0001-8615-6409

## Additional files

### Major datasets

The following dataset was generated:

| Author(s) | Year | Dataset title | Dataset URL | Database, license, and accessibility information |
|---|---|---|---|---|
| Shurtleff M, Karfilis K, Temoche-Diaz M, Ri S, Schekman R | 2016 | Data from: Y-box protein 1 is required to sort microRNAs into exosomes in cells and in a cell-free reaction | http://dx.doi.org/10.5061/dryad.83420 | Available at Dryad Digital Repository under a CC0 Public Domain Dedication |

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
