## [Decision Letter]

Thank you for submitting your work entitled "Y-box protein 1 is required to sort microRNAs into exosomes in cells and in a cell-free reaction" for consideration by *eLife*. Your article has been reviewed by three peer reviewers, one of whom is a member of our Board of Reviewing Editors, and the evaluation has been overseen by Tim Nilsen (Reviewing Editor), Jim Manley (Senior Editor), and Detlef Weigel (Deputy Editor).

The reviewers including the Reviewing Editor have extensively discussed the reviews with one another, with Senior Editor Jim Manley, and with Deputy Editor Detlef Weigel. The consensus was to decline the work in its current form, because it appears that addressing the concerns raised by the reviewers would likely require substantially more work than can be accomplished in the two months that *eLife* normally strives for. However, we are interested in the area of the work in principle, and would likely reconsider a thoroughly revised manuscript, which would be treated as a new submission. The most substantial concern related to Ago protein. That Agos were not identified from the affinity purification was one concern; the other that only single stranded miRNA, rather than a miRNA-loaded Ago complex was used in the in vitro packaging reaction.

Summary:

Here, Schekman and colleagues examine the mechanism whereby specific miRNAs are sorted into exosomes using HEK293T cells as a model system. They use a rigorous method to purify CD63 positive exosomes and find that miRs 223 and 144 are highly enriched in them relative to miRNAs in the cell. They then use a cell free system that appears to mimic the sorting and using affinity purification identify proteins bound to miR 223. YBOX1 emerges as a candidate sorting protein. Then using CRISPR knockouts of YBOX1 gene they implicate it as being involved in sorting miRNAs to exosomes.

While the reviewers agreed that the work has the potential of broad relevance expected for *eLife*, they all expressed concerns about some of the data. In particular, all of the referees questioned the use of single stranded miRNAs in the in vitro packaging reactions. It is well known that mature miRNAs are obligatorily associated with an Ago protein. The fact that Agos were not identified in the affinity purification was a major concern and brought into question the relevance of the in vitro experiments. One reviewer also noted an inconsistency in the results pertaining to the inability of YBOX1 KO cell extracts to package miR 223 when the in vivo experiments demonstrate that an additional knockdown of YBox2 is required to inhibit packaging. Questions were also raised about the sequencing results regarding both the number of reads and their mapping.

*Reviewer #1:*

Here, Schekman and colleagues examine the mechanism whereby specific miRNAs are sorted into exosomes using HEK293T cells as a model system.

They use a rigorous method to purify CD63 positive exosomes and find that miRs 223 and 144 are highly enriched in them relative to miRNAs in the cell. They then use a cell free system that appears to mimic the sorting and using affinity purification identify proteins bound to miR 223. YBOX1 emerges as a candidate sorting protein. Then using CRISPR knockouts of YBOX1 gene they implicate it as being involved in sorting the miRs to exosomes.

While the paper is clearly written and the experiments seem to be carefully performed and well controlled I do have some fairly significant concerns regarding the work and its interpretation.

Major issues:

1) All mature miRNAs in the cell are complexed with an Ago protein as a necessary prerequisite for their maturation. It is clear that miR 223 goes through normal maturation since approximately 90% of miR 223 molecules in exosomes are the mature targeting strand. Accordingly it is unclear if the cell free assays are relevant since naked miR 223 is the substrate in these reactions. Much more appropriate would be reactions in which miR 223 was built up in cells (wherein exosome synthesis was blocked by inhibitor). Mixing of extracts from such cells with exosome competent extracts would be a good experiment. If such an experiment were not possible, building up of 223 in YBOX1 YBOX2 deficient cells could also be tried.

2) Building on the previous comment there seems to be an inconsistency in the latter part of the results. On one page it is stated that extracts from YBOX1 KO cells do not support packaging of miR 223. However on the next page we learn that YBOX2 is highly induced in YBOX1 knockout cells and only after siRNA knockdown of YBOX2 do the cells become incompetent for miR 223 packaging. Please reconcile these findings.

3) It would seem appropriate to show that YBOX proteins can selectively bind miR 223 in its Ago complexed form from a pool of miRNAs. This could be done by transfecting cells with an appropriate precursor of mature miR 223 saturating the exosome producing apparatus.

*Reviewer #2:*

In this manuscript, Shurtleff and colleagues used a biochemical approach to purify an exosome species marked by CD63 and analyzed the miRNA population within the exosome. They also established a cell-free reaction to recapitulate the exosome formation process with a focus on the sorting mechanism for miRNA cargoes. They further identified YBX1 as a key molecule required for sorting miR-223/144 into the exosome. Overall, this is a very interesting and well executed study that establishes a much needed biochemical tool for studying the exosome and understanding the mechanism of cargo sorting. I have a few points that should be addressed by the authors.

1) The authors used small RNA sequencing to examine the miRNA profiles in the 293T cells and the purified exosome. However, the reads number was very low (58,848 total miRNA reads for the exosome and 511,555 total miRNAs for the cells). With the HiSeq platform, a single lane should yield at least 200M reads. Because the authors didn't provide the mapping information including how many total reads were generated, how many reads were mapped to miRNAs vs other RNAs, it is difficult to evaluate the quality of small RNA sequencing experiments. The very low reads number could be simply due to low coverage of the sequencing but could also be due to technical issues. Because the preferential loading of miR-223/144, which are lowly expressed by the cells but highly represented in the exosome, onto the exosome is a key finding of this study, it is important for the authors to provide more information for both exosome miRNA profiles and cellular miRNA profiles. Ideally, these experiments should be repeated or at least re-sequenced to generate deeper coverage.

2) To validate the biochemical purification scheme, the authors provided EM pictures to document the morphology and purity of the isolated exosome. The authors should analyze their exosome generated in their cell-free system with the same approach. Ideally, they should image the cytosol and membranes before reconstitution and the exosome after reconstitution to confirm the formation of exosome.

3) The authors used synthetic miR-223 to validate the sorting of this miRNA during cell-free exosome formation and to purify miR-223 associated proteins including YBX1. However, they used a single-stranded miR-223 for both experiments (based on the description in the method section). But I am not sure if this is the right approach. During miRNA biogenesis, one strand of the double-stranded miRNA duplex (post Dicer cleavage) is incorporated into an Ago protein. The incorporation of a single-stranded miRNA into Ago protein, which triggers a conformational change of the Ago protein according to multiple structural studies, is clearly important for the stability of both miRNA and Ago. In Dicer cKO when all miRNAs are abolished, Ago protein levels are strongly depleted and in Ago1/2 double KO most miRNAs are depleted. Therefore, it is unclear how much single-stranded, free miRNAs are available in the cytosol for sorting into the exosome. In addition, in the miR-223 associated proteome, the authors didn't identify Ago proteins. This result suggests that single-stranded miRNA can't be incorporated into Ago. The authors may consider to use miR-223 duplex for both experiments and determine how much miR-223 is sorted into the exosome and analyze whether YBX1 is still the main protein associated with the exosome miRNA.

*Reviewer #3:*

In the article by Shurtleff et al. the RNA binding protein YBX1 is implicated in driving miRNAs (specifically miR-223 and miR-144) into exosomal compartments. This is demonstrated through a combination of cell culture experiments and reconstitution assays. A highlight of the article is that several experimental approaches are pioneered. The authors developed a method to purify exosomes from cell culture media using density gradients coupled to immunoisolation via CD63 antigen. Small RNA content in the isolates is then assessed through high-throughput sequencing, revealing a set of enriched miRNAs. The two most abundantly recovered miRNA species (miR-223 and miR-144) are validated for exosome loading through qPCR and biochemical assays. The sorting of miR-223 into exosomes is then reproduced in a cell-free assay, which is subsequently used as a system for identifying factors involved in the process-with the most highly recovered factor being YBX1. Finally, the authors demonstrate the dependency of miR-223 and miR-144 sorting on YBX1 using mutant cells.

The article provides strong evidence that YBX1 is responsible for sorting of miR-223 and miR-144 into exosomes in HEK cells. As mentioned above, this is accomplished through novel purification schemes that are satisfactorily validated. Furthermore, this is an exciting area of research into a phenomena that may lead to new clinical diagnostics. Thus the article is of sufficient quality and broad interest to be suitable for publication in *eLife*. However, I feel there are a two additional experiments that if performed would significantly enhance the insights provided by the article into the functional consequences of sorting miRNAs into exosomes.

1) The authors discuss the types of miRNA species sorted into exosomes in Figure 2, which they subdivide into targeting, passenger, and precursor only. It is not clear what the authors mean by precursor only. Does this mean loop (products of dicer cleavage) or reads that map haphazardly to precursor hairpins? This is an important distinction. miRNA loops can become loaded and act as regulators in RISC. If loop species are what is meant by "precursor only" then shuttling of miRNAs may happen after loading into Argonautes. On the other hand if the recovered precursor species are derived from fragmented precursor then sorting of miRNAs may happen at the level of dicer/dicer loading complex before RISC formation. This has a significant implication: If miRNAs are diverted before Ago loading, exosome sorting may have an antagonistic role on miRNA function.

2) To further address the functional consequence of miRNA sorting into exosomes the authors should perform reporter assays for miR-223 and miR-144 mediated regulation in YBX1 mutant cells. If not sorted into exosomes do miRNAs show a gain in regulatory function? Is exosome sorting a mechanism for eliminating miRNAs from regulatory complexes? This will also provide insight into outstanding questions regarding the role of multivesicular bodies in miRNA regulation touched on in the discussion.

[Editors’ note: what now follows is the decision letter after the authors submitted for further consideration.]

Thank you for submitting your work entitled "Y-box protein 1 is required to sort microRNAs into exosomes in cells and in a cell-free reaction" for further consideration at *eLife*. Your article has been favorably evaluated by Jim Manley (Senior Editor), Timothy Nilsen (Reviewing Editor and reviewer), and two other reviewers.

The manuscript has been improved from the original submission but there are some remaining issues that need to be addressed before acceptance. All thought that the paper was significantly improved. One lingering question raised by two reviewers is whether miR-223 ever is correctly loaded by Ago proteins. Two complimentary approaches are suggested to answer this question. Please perform these analyses. Whatever the outcome the paper is now viewed to be appropriate for *eLife*, assuming the minor revisions are incorporated. You can see the reviewers' specific comments below.

*Reviewer #1:*

In the revised manuscript, Shurtleff et al., has nicely addressed several issues raised during the initial review. In particular, they used single- or double-stranded miR-223 mimics to examine their incorporation to Ago2 in the cytosol and to test selective incorporation into exosome. They showed that whereas miR-223 can be correctly programmed with Ago2, Ago2 is not detectable in the exosome. Interestingly, duplex miR-223 has higher efficiency of incorporating into the exosome than the single-stranded counterpart. In addition, the "correct" guide-strand is incorporated into the exosome, implying the same strand sorting mechanism also works for the exosome. Together with the results that Ybx1 is required for incorporating miR-223 into the exosome, these data provide strong evidence that Ybx1 but not Ago2 is associated with miRNAs in the exosome. Overall, this revised manuscript is greatly improved and should be suitable for *eLife*.

I only have a few minor comments to enhance clarity of the paper.

1) The author should use either "microRNA" or "miRNA" but not both in the manuscript.

2) The paragraph between (Results section “Exosome biogenesis in vitro) is out of order as it discusses results from Figure 4. This was before the results from Figure 4 was discussed in the following paragraph.

3) For clarity, Temp information should be provided within the panel of Figure 4, instead of using the information from Figure 4.

*Reviewer #2:*

In the resubmission of the article: "Y-box protein 1 is required to sort microRNAs into exosomes in cells and in a cell-free reaction", Shurtleff et al. added multiple experiments to explore the involvement of Ago2 in YBX1-mediated sorting of miRNAs into exosomes. These efforts showed that Ago2 is not interacting with RNAs localized to exosomes or with YBX1. Furthermore, they show duplex RNA to be more effectively sorted by YBX1 into exosomes. Together this provides some insight into the interaction of the miRNA biogenesis pathway and exosome sorting. The experiments do not distinguish, however, whether miRNAs associate with YBX1 before RISC loading or after displacement from Ago.

In my previous review I suggested authors inspect closely the alignment of small RNA reads in the exosome library to pre-miRNA hairpins. This has become a substantially more useful experiment in light of the Ago2 results. Properly processed hairpins will show precise cleavage patterns at documented Dicer cleavage sites. Clear guide, passenger, and loop reads can be recovered-all of which are competent for Ago loading. Do the exosome derived RNAs show the expected pattern? Indeed, the identity of the increased "precursor" reads in exosome libraries may provide insight into where diversion of miRNAs into exosomes occurs during their biogenesis. If precursor reads are not composed primarily of loop sequences, but rather reads that span Dicer cleavage sites this strongly suggests exosome sorting occurs before Ago loading, possibly in response to abortive Dicing. This may also help to explain why double stranded RNAs are more effectively bound by YBX1. At a minimum read alignments should be done for miR-233 and miR-144 to demonstrate the exosome sorted RNAs are appropriately processed.

Beyond this, I feel the manuscript is much improved. With this additional, easy test I support publication.

*Reviewer #3:*

The authors have addressed the major concern regarding their initial submission; i.e. the apparent absence of an Ago protein in their pull down experiments. It seems that Ago must be displaced by YBOX1 during in vitro and in vivo packaging. Alternatively, biogenesis of miR-223 may bypass Ago involvement altogether which would be a remarkable result. To distinguish between these possibilities one additional experiment would be informative. In YBOX1 knockout cells miRs 223 and 144 accumulate inside of cells. Are these miRNAs associated with Ago? A simple IP and qPCR would answer this question. Subsequent work beyond the scope of this manuscript could then sort out the biochemistry of how YBOX1 could dissociate a tightly bound miRNA from Ago. Providing this experiment is done the paper is viewed to be now acceptable for publication in *eLife*.

---

## [Author Response]

[Editors’ note: the author responses to the first round of peer review follow.]

Reviewer #1:

*Major issues:*

*1) All mature miRNAs in the cell are complexed with an Ago protein as a necessary prerequisite for their maturation. It is clear that miR 223 goes through normal maturation since approximately 90% of miR 223 molecules in exosomes are the mature targeting strand. Accordingly it is unclear if the cell free assays are relevant since naked miR 223 is the substrate in these reactions. Much more appropriate would be reactions in which miR 223 was built up in cells (wherein exosome synthesis was blocked by inhibitor). Mixing of extracts from such cells with exosome competent extracts would be a good experiment. If such an experiment were not possible, building up of 223 in YBOX1 YBOX2 deficient cells could also be tried.*

These experiments are challenging because of the paucity of data regarding exosome biogenesis. Accordingly, no specific inhibitor of exosome biogenesis has yet been discovered. We addressed the Ago concerns directly in the new Figure 6. Further, we show in Figure 7 that miR-223 and miR-144 accumulate within YBX1 KO cells.

*2) Building on the previous comment there seems to be an inconsistency in the latter part of the results. On one page it is stated that extracts from YBOX1 KO cells do not support packaging of miR 223. However on the next page we learn that YBOX2 is highly induced in YBOX1 knockout cells and only after siRNA knockdown of YBOX2 do the cells become incompetent for miR 223 packaging. Please reconcile these findings.*

The YBX-paralog expression data has been removed from the revised manuscript. The secretion of miR-223 is significantly decreased in YBX1-KO cells but is further diminished after YBX2 knockdown. We believe this result may be explained by partial functional redundancy for YBX2 with respect to miR-223 but not for miR-144 that might depend on differences in binding specificities between the two paralogs. We added this explanation to the revised manuscript. In addition, it is worth emphasizing that the two assays, ss RNA incorporation a cell-free reaction and RNA sorting into exosomes secreted by intact cells, are measured over quite different time scales: 20 min vs. one day. Thus, a rate difference in the two reactions may be difficult to interpret.

*3) It would seem appropriate to show that YBOX proteins can selectively bind miR 223 in its Ago complexed form from a pool of miRNAs. This could be done by transfecting cells with an appropriate precursor of mature miR 223 saturating the exosome producing apparatus.*

We are again constrained by the lack of knowledge on the exosome biogenesis machinery. It is often assumed that the machinery involved in exosome biogenesis is the same as the MVB machinery, a primary degradative pathway within the cell. It is not clear that over-expressing a single miRNA precursor could saturate the exosome producing apparatus. However, we believe the basis of this concern is addressed in the new Figure 6. While ss and duplex synthetic miRNA sequences can be bound by Ago2 in the cytosol only fractions (c) in the complete reaction Ago2 is undetectable on the sequestered ss RNA whereas YBX1 is bound at levels that mirror the packaging efficiency (d,e,f).

Reviewer #2:

*In this manuscript, Shurtleff and colleagues used a biochemical approach to purify an exosome species marked by CD63 and analyzed the miRNA population within the exosome. They also established a cell-free reaction to recapitulate the exosome formation process with a focus on the sorting mechanism for miRNA cargoes. They further identified YBX1 as a key molecule required for sorting miR-223/144 into the exosome. Overall, this is a very interesting and well executed study that establishes a much needed biochemical tool for studying the exosome and understanding the mechanism of cargo sorting. I have a few points that should be addressed by the authors.*

*1) The authors used small RNA sequencing to examine the miRNA profiles in the 293T cells and the purified exosome. However, the reads number was very low (58,848 total miRNA reads for the exosome and 511,555 total miRNAs for the cells). With the HiSeq platform, a single lane should yield at least 200M reads. Because the authors didn't provide the mapping information including how many total reads were generated, how many reads were mapped to miRNAs vs other RNAs, it is difficult to evaluate the quality of small RNA sequencing experiments. The very low reads number could be simply due to low coverage of the sequencing but could also be due to technical issues. Because the preferential loading of miR-223/144, which are lowly expressed by the cells but highly represented in the exosome, onto the exosome is a key finding of this study, it is important for the authors to provide more information for both exosome miRNA profiles and cellular miRNA profiles. Ideally, these experiments should be repeated or at least re-sequenced to generate deeper coverage.*

We have re-analzyed these datasets and provided the mapping details in the new Figure 2—supplementary table 1. In this study we used the small RNA-seq as an initial screening method to identify candidate miRNAs that may be highly enriched in exosomes. We believe that the strongest evidence for miR-223 and miR-144 as true exosomal miRNAs is presented in Figure 2 which shows that the miRNAs are enriched at each stage of the biochemical purification and are inside of detergent sensitive vesicles. Since we did not follow up on more miRNAs in this study we have removed from the manuscript the description of highly enriched vs. moderately enriched miRNAs so as to not overinterpret our data.

*2) To validate the biochemical purification scheme, the authors provided EM pictures to document the morphology and purity of the isolated exosome. The authors should analyze their exosome generated in their cell-free system with the same approach. Ideally, they should image the cytosol and membranes before reconstitution and the exosome after reconstitution to confirm the formation of exosome.*

The requested experiment is technically challenging. The input membrane fraction for the in vitro reaction is a crude sample containing all cellular membranes that sediment at 15,000Xg. This will include many different structures, including vesicles, membrane sheets as well as non-membranous structures. The formation of vesicles by EM pre and post incubation is unlikely to show any difference given the large excess of material in the reconstitution reaction. Furthermore, any vesicles that are seen might have been budded from other organelles present in the sample (e.g. ER or Golgi). Finally, it is not expected that the reconstitution forms free vesicles, but rather intralumenal vesicles within multivesicular bodies. These newly formed vesicles would not be visible using the simple negative staining EM presented in Figure 1.

*3) The authors used synthetic miR-223 to validate the sorting of this miRNA during cell-free exosome formation and to purify miR-223 associated proteins including YBX1. However, they used a single-stranded miR-223 for both experiments (based on the description in the method section). But I am not sure if this is the right approach. During miRNA biogenesis, one strand of the double-stranded miRNA duplex (post Dicer cleavage) is incorporated into an Ago protein. The incorporation of a single-stranded miRNA into Ago protein, which triggers a conformational change of the Ago protein according to multiple structural studies, is clearly important for the stability of both miRNA and Ago. In Dicer cKO when all miRNAs are abolished, Ago protein levels are strongly depleted and in Ago1/2 double KO most miRNAs are depleted. Therefore, it is unclear how much single-stranded, free miRNAs are available in the cytosol for sorting into the exosome. In addition, in the miR-223 associated proteome, the authors didn't identify Ago proteins. This result suggests that single-stranded miRNA can't be incorporated into Ago. The authors may consider to use miR-223 duplex for both experiments and determine how much miR-223 is sorted into the exosome and analyze whether YBX1 is still the main protein associated with the exosome miRNA.*

These concerns are addressed in the revised manuscript and the new Figure 6.

Reviewer #3:

*[…] The article provides strong evidence that YBX1 is responsible for sorting of miR-223 and miR-144 into exosomes in HEK cells. As mentioned above, this is accomplished through novel purification schemes that are satisfactorily validated. Furthermore, this is an exciting area of research into a phenomena that may lead to new clinical diagnostics. Thus the article is of sufficient quality and broad interest to be suitable for publication in eLife. However, I feel there are a two additional experiments that if performed would significantly enhance the insights provided by the article into the functional consequences of sorting miRNAs into exosomes.*

*1) The authors discuss the types of miRNA species sorted into exosomes in Figure 2, which they subdivide into targeting, passenger, and precursor only. It is not clear what the authors mean by precursor only. Does this mean loop (products of dicer cleavage) or reads that map haphazardly to precursor hairpins? This is an important distinction. miRNA loops can become loaded and act as regulators in RISC. If loop species are what is meant by "precursor only" then shuttling of miRNAs may happen after loading into Argonautes. On the other hand if the recovered precursor species are derived from fragmented precursor then sorting of miRNAs may happen at the level of dicer/dicer loading complex before RISC formation. This has a significant implication: If miRNAs are diverted before Ago loading, exosome sorting may have an antagonistic role on miRNA function.*

We thank the reviewer for pointing out our error in terminology in Figure 2. This should have been denoted precursor. This category was derived from the quantifier program of the miRdeep2 package. Briefly, reads were mapped to all precursor miRNAs and mature miRNAs separately. The number of reads mapping to the mature were then subtracted from the reads mapping to the precursor to identify sequences that map only to the precursor and thus likely arise from a precursor molecule. Since the vast majority of miRNA reads mapped to the mature sequences, we did not follow up in detail as to whether the precursor reads are likely full-length or if they are derived from fragments.

*2) To further address the functional consequence of miRNA sorting into exosomes the authors should perform reporter assays for miR-223 and miR-144 mediated regulation in YBX1 mutant cells. If not sorted into exosomes do miRNAs show a gain in regulatory function? Is exosome sorting a mechanism for eliminating miRNAs from regulatory complexes? This will also provide insight into outstanding questions regarding the role of multivesicular bodies in miRNA regulation touched on in the discussion.*

These are interesting questions but are beyond the scope on this manuscript in which we describe a mechanism by which miRNAs are selectively sorted into exosomes. Experiments are currently underway to understand the function and cellular importance of miRNA sorting into exosomes.

[Editors' note: the author responses to the re-review follow.]

Reviewer #1:

*[…] I only have a few minor comments to enhance clarity of the paper.*

*1) The author should use either "microRNA" or "miRNA" but not both in the manuscript.*

*2) The paragraph between (Results section “Exosome biogenesis in vitro) is out of order as it discusses results from Figure 4. This was before the results from Figure 4 was discussed in the following paragraph.*

*3) For clarity, Temp information should be provided within the panel of Figure 4, instead of using the information from Figure 4.*

We thank the reviewer for these important comments regarding manuscript clarity.

1) miRNA has been substituted for all instances of microRNA except in the first usage which has been edited to read "microRNA (miRNA)".

2) The text in the biogenesis section referring to Figure 4 has been moved to the next section and is now discussed in the order presented in the figures.

3) Thank you for pointing out this error in the Figure 4. We have now included the Temp information under the axis.

Reviewer #2:

*[…] In my previous review I suggested authors inspect closely the alignment of small RNA reads in the exosome library to pre-miRNA hairpins. This has become a substantially more useful experiment in light of the Ago2 results. Properly processed hairpins will show precise cleavage patterns at documented Dicer cleavage sites. Clear guide, passenger, and loop reads can be recovered-all of which are competent for Ago loading. Do the exosome derived RNAs show the expected pattern? Indeed, the identity of the increased "precursor" reads in exosome libraries may provide insight into where diversion of miRNAs into exosomes occurs during their biogenesis. If precursor reads are not composed primarily of loop sequences, but rather reads that span Dicer cleavage sites this strongly suggests exosome sorting occurs before Ago loading, possibly in response to abortive Dicing. This may also help to explain why double stranded RNAs are more effectively bound by YBX1. At a minimum read alignments should be done for miR-233 and miR-144 to demonstrate the exosome sorted RNAs are appropriately processed.*

*Beyond this, I feel the manuscript is much improved. With this additional, easy test I support publication.*

We have addressed the reviewer's concern by performing a read frequency distribution analysis for miR-223 and miR-144 along the length of the hairpin precursors. These results indicate that the vast majority of reads map to the mature guide strand of each miRNA with few reads mapping to passenger strand. We saw no evidence of aberrant Dicer cleavage products in this analysis. This data has been added as a Figure 2—figure supplement 1 and the following text has been added to the Results section describing miR-223 and miR-144 enrichment in exosomes:

"An analysis of the read frequency distribution of miR-223 and miR-144 from the exosome small RNA-seq dataset showed that the vast majority of reads mapped to the mature guide strand with few reads also mapping to the passenger strand (Figure 2—figure supplement 1)."

Reviewer #3:

*The authors have addressed the major concern regarding their initial submission; i.e. the apparent absence of an Ago protein in their pull down experiments. It seems that Ago must be displaced by YBOX1 during* in vitro *and* in vivo *packaging. Alternatively, biogenesis of miR-223 may bypass Ago involvement altogether which would be a remarkable result. To distinguish between these possibilities one additional experiment would be informative. In YBOX1 knockout cells miRs 223 and 144 accumulate inside of cells. Are these miRNAs associated with Ago? A simple IP and qPCR would answer this question. Subsequent work beyond the scope of this manuscript could then sort out the biochemistry of how YBOX1 could dissociate a tightly bound miRNA from Ago. Providing this experiment is done the paper is viewed to be now acceptable for publication in eLife.*

The reviewer raises an important question concerning an association of miR-223 and Ago in YBX1 mutant cells. We performed the requested experiment by IPing Ago in WT and ∆YBX1 cells and detected bound miR-223 by qPCR. We found no detectable miR-223 above background associated with Ago2 in WT cells, however, in KO cells, we detected miR-223 in complex with Ago2. These results support the notion that miR-223 undergoes normal biogenesis and may be loaded into Ago2 and then, in HEK293T wild type cells, is efficiently displaced or simply replaced by YBX1. Future work will determine if YBX1 is directly involved in the dissociation of miR-223 from Ago2 or simply binds miR-223 that is released from Ago2 via another mechanism.

The figure below was added to the Figure 7—figure supplement 2 and the following text was inserted in the final paragraph of the Results section:

“To determine if Ago2 binds miR-223 that accumulates in ∆YBX1 cells, we performed RNA immunoprecipitation with Ago2 or isotype matched control IgG in lysates of WT and ∆YBX1 cells and quantified the amount of associated miR-223 by qPCR (Figure 7—figure supplement 2). No miR-223 above background co-immunoprecipitated with Ago2 in WT cells. In contrast, miR-223 accumulated in ∆YBX1 cells was found to be associated with Ago2. These results suggest that miR-223 is properly processed in HEK293T cells, likely loaded into Ago2 and then efficiently dissociated and bound by YBX1.”